# A Signed Graph Approach to Understanding and Mitigating Oversmoothing

Jiaqi Wang[1*]    Xinyi Wu[2*]    James Cheng[1]    Yifei Wang[3†]

[1] The Chinese University of Hong Kong    [2]MIT IDSS & LIDS    [3]MIT CSAIL

{jqwang23, jcheng}@cse.cuhk.edu.hk    {xinyiwu, yifei_w}@mit.edu

## Abstract

Deep graph neural networks (GNNs) often suffer from oversmoothing, where node representations become overly homogeneous with increasing depth. While techniques like normalization, residual connections, and edge dropout have been proposed to mitigate oversmoothing, they are typically developed independently, with limited theoretical understanding of their underlying mechanisms. In this work, we present a unified theoretical perspective based on the framework of signed graphs, showing that many existing strategies implicitly introduce negative edges that alter message-passing to resist oversmoothing. However, we show that merely adding negative edges in an unstructured manner is insufficient—the asymptotic behavior of signed propagation depends critically on the strength and organization of positive and negative edges. To address this limitation, we leverage the theory of structural balance, which promotes stable, cluster-preserving dynamics by connecting similar nodes with positive edges and dissimilar ones with negative edges. We propose Structural Balanced Propagation (SBP), a plug-and-play method that assigns signed edges based on either labels or feature similarity to explicitly enhance structural balance in the constructed signed graphs. Experiments on nine benchmarks across both homophilic and heterophilic settings demonstrate that SBP consistently improves classification accuracy and mitigates oversmoothing, even at depths of up to 300 layers. Our results provide a principled explanation for prior oversmoothing remedies and introduce a new direction for signed message-passing design in deep GNNs. Our code is available at `https://github.com/kokolerk/sbp`.

## 1 Introduction

Graph neural networks (GNNs) are a powerful framework for processing graph-structured data from diverse application domains [1, 2, 3, 4, 5, 6, 7]. Most GNN models follow the *message-passing* paradigm, where node representations are computed by recursively aggregating information from neighboring nodes along the edges [8, 9, 10, 11]. Despite their empirical success, deep GNNs often suffer from oversmoothing—the tendency of node features to become indistinguishable as layers increase—leading to performance degradation in deeper models [12, 13, 14, 15].

Numerous techniques have been proposed to mitigate oversmoothing in GNNs, including normalization layers [16, 17], residual connections [18, 19, 20], and random edge dropout [21, 22]. While empirically effective, these methods are typically developed independently, with limited theoretical understanding of the mechanisms that underlie their success. A common challenge is that many of them introduce architectural modifications that alter the message-passing process [23], making it difficult to precisely characterize their effects on propagation dynamics and the resulting node

---

[*]Equal contribution.

[†]Corresponding authors.

39th Conference on Neural Information Processing Systems (NeurIPS 2025).

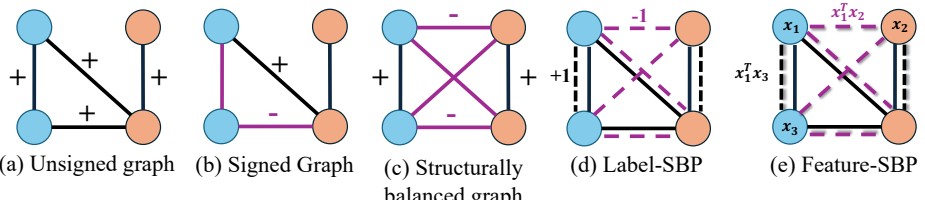

Figure 1: Examples of signed graph structures. Blue and orange circles represent nodes from different classes. Solid lines denote real edges, while dashed lines represent edges introduced by SBP. Black and purple lines indicate positive and negative edges, respectively. Let $x_i$ be the node features for node $i$. (a) Initial unsigned graph. (b) Signed graph. (c) Ideal structurally balanced graph. (d),(e) Graphs induced by Label-SBP and Feature-SBP, respectively.

representations. Moreover, their ability to prevent oversmoothing is often limited, especially in deep GNNs, where they are observed to fail to preserve discriminative node features at extreme propagation depths [16, 20, 24].

In this work, we present a unified theoretical perspective on oversmoothing in GNNs, showing that many existing mitigation techniques can be interpreted as implicitly introducing negative edges into the graph used for message-passing. We formalize this insight using the framework of signed graphs [25], where edges carry either positive or negative signs (Figure 1(b)). In this view, positive edges promote alignment, while negative edges introduce repulsion, shaping the long-term dynamics of node features under signed propagation. However, we further show that simply adding negative edges in an unstructured manner is insufficient, as the asymptotic behavior of signed message-passing depends not only on the presence of negative edges but also on the strength and organization of positive and negative edges. To address this, we turn to the theory of *structural balance* [26], which characterizes graphs where positive edges connect nodes within clusters and negative edges connect nodes across clusters (Figure 1(c)). We prove that message-passing on such graphs yields stable, cluster-preserving dynamics, preventing oversmoothing while enhancing class separation.

Motivated by this theory, we propose **S**tructural **B**alance **P**ropagation (SBP), a simple, plug-and-play module without introducing learnable parameters for constructing signed graphs that promote structural balance. SBP comes in two variants: (1) Label-SBP, which assigns signs based on ground-truth labels (Figure 1(d)), and (2) Feature-SBP, which estimates signs from feature similarity for label-scarce settings (Figure 1(e)). We theoretically show that Label-SBP induces structural balance under mild conditions. Empirically, we evaluate both variants on nine synthetic and real-world benchmarks across homophilic and heterophilic settings. Our results show that SBP consistently improves classification performance and mitigates oversmoothing, validating our theoretical findings. Finally, we analyze the robustness of SBP to design choices, highlighting its adaptability and reliability across diverse GNN settings.

**Our main contributions are summarized as follows:**

- We provide a unified theoretical perspective showing that many oversmoothing mitigation techniques such as normalization, residual connections, and edge dropout, can be interpreted as implicitly introducing negative edges into the graph. We formalize this insight through the framework of signed graphs and further show that the asymptotic behavior of signed propagation depends critically on the strength and organization of both positive and negative edges.

- We identify structural balance as an ideal condition for resisting oversmoothing, proving that it guarantees stable, class-distinct representations under signed message-passing.

- Based on this theory, we propose Structural Balanced Propagation (SBP), a simple, plug-and-play method that constructs signed graphs designed to promote structural balance, using either label information (Label-SBP) or feature similarity (Feature-SBP).

- Extensive experiments on nine benchmarks demonstrate that SBP consistently improves classification and mitigates oversmoothing across both homophilic and heterophilic graphs. Our analysis also highlights the method's robustness and adaptability to different design choices.

## 2 Related Work

**Theory of Oversmoothing.** The notion of oversmoothing was first introduced by [12], who observed that node representations tend to converge to a common value as GNN depth increases. Subsequent work [13, 15] provided rigorous proofs showing that this convergence occurs at an exponential rate for GCNs and attention-based GNNs. [24] further showed that oversmoothing can arise even in shallow networks under specific random graph models. [23] proved that residual connections and normalization layers can mitigate oversmoothing, but they introduce their own limitations by altering the original message-passing process.

**Signed Graph-Inspired Methods.** Several methods have drawn inspiration from signed graph propagation to handle heterophilic graphs [27, 28, 29, 30, 31], where vanilla GNNs tend to perform worse than on homophilic graphs. [29, 30] leveraged negative edges to encode dissimilarity and introduce repulsion in message-passing. [31] made layer aggregation coefficients learnable and observed that negative edges naturally emerge in heterophilic settings. However, [32] showed that oversmoothing can still occur in signed propagation under certain random graph models, suggesting that simply adding negative edges is not sufficient to guarantee expressive representations. This highlights a broader limitation of existing approaches, which often rely on heuristic designs without a principled understanding of when and why signed message-passing is effective. In this work, we provide a theoretical characterization of how the strength and structure of negative edges influence the asymptotic behavior of node representations, and we propose Structural Balanced Propagation (SBP) to promote stable and discriminative representations on general graphs.

## 3 How Oversmoothing Happen in GNNs? A Signed Graph Perspective

In this section, we present a generalized message-passing framework based on signed graphs, by incorporating both attractive (positive) and repulsive (negative) interactions. We show that many oversmoothing mitigation techniques can be reinterpreted as implicitly introducing negative edges. However, we demonstrate that presence of negative edges is not sufficient—the asymptotic behavior of signed propagation depends critically on the strength of positive and negative edges.

**Signed Graphs.** We represent an unsigned, undirected graph with $n$ nodes by $\mathcal{G} = (A, X)$, where $A \in \{0, 1\}^{n \times n}$ denotes the adjacency matrix and $X \in \mathbb{R}^{n \times d}$ is the node feature matrix. For node $i, j \in \{1, 2, .., n\}$, $A_{i,j} = 1$ if and only if node $i, j$ are connected by an edge in $\mathcal{G}$ and $X_i \in \mathbb{R}^d$ represents the features of node $i$. We let $\mathbb{1}_n$ be the all-one vector of length $n$ and $D = \text{diag}(A\mathbb{1}_n)$ be the degree matrix of $\mathcal{G}$.

A signed graph associates each edge with a positive or negative sign, capturing the notion of attraction or repulsion between nodes. In this paper, we extend $\mathcal{G}$ to the signed graph $\mathcal{G}_s = \{A^+, A^-, X\}$ where $A^+, A^- \in \{0, 1\}^{n \times n}$ are the positive and negative adjacency matrices capturing positive and negative edges, with the degree matrix $D_+ = \text{diag}(A^+\mathbb{1}_n)$ and $D_- = \text{diag}(A^-\mathbb{1}_n)$, respectively.

**Signed Graph Propagation.** Following [25, 33], we define the signed propagation which happen over both positive and negative edges with neighboring nodes [25] as

$$X_i^{(k)} = (1 - \alpha + \beta)X_i^{(k-1)} + \frac{\alpha}{D_i^+} \sum_{j \in N_i^+} X_j^{(k-1)} - \frac{\beta}{D_i^-} \sum_{j \in N_i^-} X_j^{(k-1)}, \tag{1}$$

where $N_i^+$ and $N_i^-$ represent the set of positive and negative neighbors for node $i$, $D_i^+$ and $D_i^-$ represent the positive and negative degrees for node $i$, respectively.

To allow for a general formulation, we introduce two hyperparameters: $\alpha, \beta > 0$, which control the strength of the propagation over the positive and negative edges, respectively. In particular, when $\beta = 0$ and $\alpha = 1$, (1) would correspond to the unsigned graph propagation $X_i^{(k)} = \frac{1}{D_i^+} \sum_{j \in N_i^+} X_j^{(k-1)}$ in vanilla message-passing.

**Prior Methods from the Lens of Signed Graphs.** Many previously proposed oversmoothing mitigation techniques can be reinterpreted as special cases of the signed graph propagation in (1). In particular, we observe the following:

**Proposition 3.1** *Normalization layers, residual connections, and random edge dropout can all be expressed as instances of signed graph propagation in (1), where the vanilla unsigned message-passing*

*is modified by implicitly injecting non-trivial negative edges. A summary of these correspondences is provided in Table 5 in Appendix E.*

To isolate the effect of signed propagation on oversmoothing, we focus our theoretical analysis on the linear setting by removing the activation function and setting $\sigma(x) = x$, as in prior works [9, 24]. In the following theorem, we formally characterize how the strength of negative edges governs the long-term behavior of node representations.

**Theorem 3.2** *Suppose that in a signed graph $\mathcal{G}_s$, where $A^+$ represents a connected graph and $X_i^{(k)}$ represents the value of node $i$ after $k$ propagation steps under (1). Then for any $0 < \alpha < 1/\max_{i \in X} D_i^+$, there exists a critical value $\beta_* \geq 0$ such that:*

*(i)  if $\beta < \beta_*$, then we have $\lim_{k \to \infty} X_i^{(k)} = \sum_{j=1}^{n} X_j^{(0)}/n$ for all initial values $X_i^{(0)}$;*

*(ii)  if $\beta > \beta_*$, then $\lim_{k \to \infty} \|X^{(k)}\| = \infty$ for almost all initial values w.r.t. Lebesgue measure.*

The proof is provided in Appendix C. Theorem 3.2 highlights the pivotal role of the negative edge weight $\beta$: it acts as a repulsive force that counterbalances the homogenizing effect of positive-edge aggregation. When $\beta$ is small, especially in the extreme case where $\beta = 0$, negative edges vanish from the dynamics, and the model degenerates into standard unsigned propagation, leading to inevitable oversmoothing, regardless of the choice of $\alpha$. Crucially, although increasing $\beta$ can prevent oversmoothing by preserving heterogeneity in node representations, excessively strong repulsion causes the dynamics to become unstable, with representations diverging toward infinity. This tradeoff poses a challenge: how can we retain the benefits of negative edges to mitigate oversmoothing without destabilizing the model?

To address this, we turn to the theory of structural balance, which characterizes configurations of signed graphs where the tension between positive and negative edges is globally well-structured.

# 4  Our Proposal: Structural Balance Propagation

In this section, we propose that message-passing over *structurally balanced* signed graphs exhibit controllable and stable asymptotic behavior, making them theoretically well-suited for mitigating oversmoothing in deep GNNs. Building on this insight, we introduce Structural Balanced Propagation (SBP), a simple and effective approach that explicitly promotes structural balance in the constructed signed graph.

## 4.1  Asymptotic Behavior of Propagation over Structurally Balanced Graphs

In the previous section, we demonstrated that oversmoothing arises when the influence of negative edges is insufficient to counteract the homogenizing effect of positive-edge propagation. Conversely, overly strong negative edges lead to divergence and instability. This reveals a fundamental tension in signed message-passing: to avoid oversmoothing while ensuring stability, the distribution and strength of signed edges must be carefully controlled.

To address this challenge, we turn to a special class of signed graphs known as structurally balanced graphs. These graphs encode an ideal configuration in which positive edges connect nodes within the same cluster, and negative edges span across clusters. Crucially, under signed propagation, such structure leads to stable asymptotic behavior that preserves intra-cluster similarity and inter-cluster distinction—precisely the property needed to resist oversmoothing and enhance classification performance in deep GNNs [24]. Formally, following [25, 26], we define structural balance as follows:

**Definition 4.1 (Structurally Balanced Graph)** *A signed graph $\mathcal{G}_s$ is called **structurally balanced** if there is a partition of the node set into $V = V_1 \cup V_2$ with $V_1$ and $V_2$ being nonempty and mutually disjoint, where any edge between the two node subsets $V_1$ and $V_2$ is negative, and any edge within each $V_i$ is positive.*

The structural balance property partitions the graph into two disjoint node sets, $V_1$ and $V_2$, such that positive edges connect nodes within the same group, while negative edges connect nodes

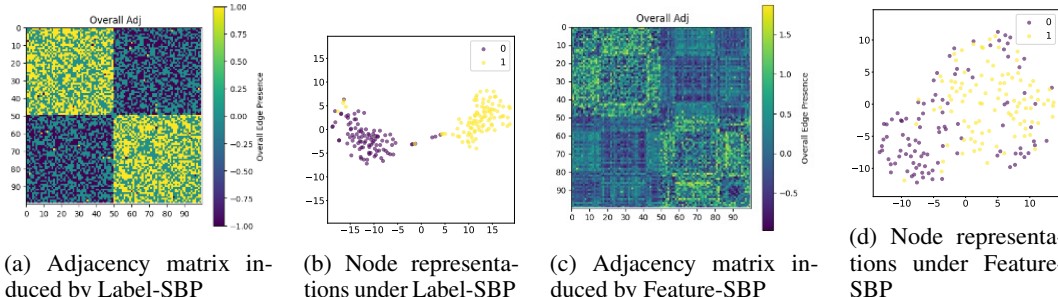

(a) Adjacency matrix induced by Label-SBP

(b) Node representations under Label-SBP

(c) Adjacency matrix induced by Feature-SBP

(d) Node representations under Feature-SBP

Figure 2: The visualization of the signed adjacency matrix $A^+ - A^-$ induced by SBP and resulting node representations on 2-CSBM under Layer= 300. (a)(c): The X-axis and Y-axis denote the nodes 0-99, where 0-49 is from class 0 and 50-99 is from class 1. (b)(d): The t-SNE visualization of the node representations learned by SBP.

across groups, as illustrated in Figure 1 (c). This organization of edge signs induces well-structured propagation dynamics. We characterize the asymptotic behavior of signed message-passing on structurally balanced graphs as follows:

**Theorem 4.2** *Assume that node $i$ is connected to node $j$ and $X_i^{(k)}$ represents the value of node $i$ after $k$ propagation steps in (1). $\mathcal{F}(z)_c$ is a bounded function such that: if $z < -c$, $\mathcal{F}(z)_c = -c$; if $z > c$, $\mathcal{F}(z)_c = c$; if $-c < z < c$, $\mathcal{F}(z)_c = z$. Let $\theta = \alpha$ if the edge $\{i, j\}$ is positive and $\theta = -\beta$ if the edge $\{i, j\}$ is negative. Consider the constrained signed propagation update:*

$$X_i^{(k+1)} = \mathcal{F}_c((1 - \theta)X_i^{(k)} + \theta X_j^{(k)}). \tag{2}$$

*Let $\alpha \in (0, 1/2)$. Assume that $\mathcal{G}_s$ is a structurally balanced complete graph under the partition $V = V_1 \cup V_2$. When $\beta$ is sufficiently large, we have that*

$$\mathbb{P}\left(\lim_{k \to \infty} X_i^{(k)} = c, i \in V_1; \lim_{k \to \infty} X_i^{(k)} = -c, i \in V_2\right) = 1. \tag{3}$$

The proof is provided in Appendix D. The result above shows that if the graph is structurally balanced and the signed graph propagation is constrained with a bounded function $\mathcal{F}_c$, node features converge asymptotically to group-specific values under the propagation rule defined in (1). Moreover, nodes belonging to different groups are repelled from one another, resulting in asymptotically distinct representations across groups. This behavior implies that structurally balanced graphs provide a provable mechanism for mitigating oversmoothing, by assigning positive and negative edge signs in accordance with the underlying class structure—encouraging intra-class consistency while maintaining inter-class separation.

**Remark 4.3** *The two-group result above can be generalized to multiple groups by introducing a more general notion known as weak structural balance. See detailed discussion in Appendix G.*

### 4.2 Method: Design Structural Balance Propagation for GNNs

Building on the theoretical insights from the previous section, we propose **S**tructural **B**alance **P**ropagation (SBP), a principled approach for promoting structural balance in the message-passing process of GNNs. Specifically, we introduce two variants: Label-SBP and Feature-SBP, which inject signed edges that approximate structurally balanced configurations using label supervision and feature similarity, respectively.

**Label Induced Structural Balance Propagation (Label-SBP).** We extend the original adjacency matrix to the positive and negative ones. Let the positive adjacency matrix be the original adjacency matrix $A^+ = A$, we then construct a label-informed negative adjacency matrix $A_l^-$, designed to introduce repulsion between classes and promote attraction within classes. Specifically, for node pairs with known labels, we assign a value of $1$ if the labels differ (to repel), $-1$ if the labels match (to attract), and $0$ otherwise to preserve the original structure when labels are unknown. Formally:

$$(A_l^-)_{ij} = \begin{cases} 1 & \text{if } y_i \neq y_j, \\ -1 & \text{if } y_i = y_j, \\ 0 & \text{otherwise}, \end{cases} \tag{4}$$

where $y_i$ is the ground truth label for node $i$. We theoretically show that Label-SBP induces a structurally balanced graph under mild conditions (see formal statement and proof in Appendix H).

**Theorem 4.4 (Informal)** *Assume class labels are balanced, and let $p$ denote the ratio of labeled nodes. As $p$ increases, the degree of structural balance improves, and the graph becomes fully structurally balanced when $p = 1$.*

Figure 2a shows the signed adjacency matrix $A^+ - A_l^-$, constructed by Label-SBP on the the Contextual Stochastic Block Model with two blocks (2-CSBM) [34], highlighting its structural balance: positive edges appear within label blocks, and negative edges span between them. Figure 2b visualizes the learned node representations, demonstrating that Label-SBP achieves clear class separation even at depth $L = 300$, attaining a high classification accuracy of 97.50%, which is consistent with Theorem 4.2.

Furthermore, to tackle scenarios where labels are scarce, we propose a variant of SBP that estimates the negative adjacency matrix based on feature similarities.

**Feature Induced Structural Balance Propagation (Feature-SBP).** We retain the positive adjacency matrix in Feature-SBP as in Label-SBP, setting $A^+ = A$. To construct the negative adjacency matrix $A_f^-$, We leverage feature similarity to assign positive and negative edges—promoting attraction between similar nodes and introducing repulsion between dissimilar ones—without relying on labels. Specifically, we define:

$$A_f^- = -X^{(0)}X^{(0)\top}, \tag{5}$$

where $X^{(0)}$ denotes the initial node features. While this approach may be less precise than Label-SBP due to the absence of label supervision, it leverages the full feature set, including test nodes, to improve the overall alignment with structural balance across the graph.

Figure 2c and 2d illustrate the signed adjacency matrix $A^+ - A_f^-$ and learned node representations on the 2-CSBM data. Notably, Feature-SBP preserves structural balance patterns similar to Label-SBP and achieves strong classification performance with an accuracy of 80.00%.

**Implementation Details.** We implement the constrained function $\mathcal{F}_c$ in Theorem 4.2 by Layer-Norm [35]. To avoid numerical instability for repeated message-passing, we ensure that the sum of the coefficients combining the node representations $X^{(k)}$ and the node representations updates by our SBP remains 1. We employ a row-stochastic adjacency matrix $\hat{A}$ as the positive adjacency matrix, denoted $\hat{A}^+$. Additionally, we apply the softmax function to the negative matrix, resulting in $\hat{A}^- = \text{softmax}(A^-)$. As a result, Label/Feature-SBP can be written as:

$$X^{(k+1)} = (1 - \lambda)X^{(k)} + \lambda\Big(\alpha\hat{A}^+X^{(k)} - \beta\hat{A}^-X^{(k)}\Big).$$

where $0 < \lambda < 1, \alpha, \beta > 0$ are the hyperparameters controlling the strength of positive and negative edges.

**Scalability on Large-Scale Graphs.** Although SBP improves the structural balance property for message-passing in GNNs, it may reduce graph sparsity and cause out-of-memory issues on large-scale graphs. To adapt SBP in large-scale graphs, we propose Label-SBP-v2, which removes only inter-class edges instead of explicitly adding negative ones. This preserves sparsity by avoiding the addition of new edges, thereby reducing computational overhead while still encouraging structural balance.

For Feature-SBP, the original negative adjacency matrix has quadratic complexity $\mathcal{O}(n^2d)$, which is prohibitive for large $n$. To improve efficiency, we replace the node-level similarity matrix $-XX^\top \in \mathbb{R}^{n \times n}$ with its transposed form $-X^\top X \in \mathbb{R}^{d \times d}$, following [16]. This shifts repulsion to the feature dimension and reduces complexity to $\mathcal{O}(nd^2)$, which is significantly more scalable when $n \gg d$. More detailed analysis is provided in Appendix I.3.

## 5   Experiments

In this section, we conduct a comprehensive evaluation of SBP on various benchmark datasets, including both homophilic and heterophilic graphs. We aim to answer the following three key research questions:

Table 2: Node classification accuracy using standard-depth GNNs (%). Best results are highlighted in blue; second-best results are shown in gray. Overall, SBP performs best on both homophilic and heterophilic datasets.

| $H(G)$ Dataset | 0.81 Cora | 0.74 Citeseer | 0.80 PubMed | 0.22 Squirrel | 0.38 Amazon-ratings | 0.21 Texas | 0.11 Wisconsin | 0.30 Cornell |
|---|---|---|---|---|---|---|---|---|
| MLP | $48.82 \pm 0.98$ | $47.89 \pm 1.21$ | $69.20 \pm 0.83$ | $32.58 \pm 0.19$ | $38.14 \pm 0.03$ | $73.51 \pm 2.36$ | $70.98 \pm 1.18$ | $68.11 \pm 2.65$ |
| SGC | $80.21 \pm 0.07$ | $71.88 \pm 0.27$ | $76.99 \pm 0.38$ | $43.30 \pm 0.30$ | $42.83 \pm 0.04$ | $45.95 \pm 0.00$ | $47.06 \pm 0.00$ | $48.11 \pm 3.15$ |
| BatchNorm | $77.90 \pm 0.00$ | $60.85 \pm 0.09$ | $77.15 \pm 0.09$ | $44.22 \pm 0.11$ | $39.68 \pm 0.01$ | $39.73 \pm 1.24$ | $52.94 \pm 0.00$ | $46.49 \pm 1.08$ |
| PairNorm | $80.30 \pm 0.05$ | $70.83 \pm 0.06$ | $77.69 \pm 0.26$ | $46.21 \pm 0.09$ | $42.30 \pm 0.05$ | $51.35 \pm 0.00$ | $58.82 \pm 0.00$ | $51.35 \pm 0.00$ |
| ContraNorm | $81.60 \pm 0.00$ | $72.25 \pm 0.08$ | $79.30 \pm 0.10$ | $48.63 \pm 0.16$ | $42.98 \pm 0.04$ | $48.38 \pm 4.43$ | $49.61 \pm 1.53$ | $48.63 \pm 0.16$ |
| DropEdge | $73.58 \pm 2.76$ | $65.63 \pm 1.76$ | $74.64 \pm 1.37$ | $42.30 \pm 0.62$ | $42.30 \pm 0.09$ | $59.46 \pm 8.11$ | $52.55 \pm 4.45$ | $45.95 \pm 7.05$ |
| Residual | $77.81 \pm 0.03$ | $71.61 \pm 0.17$ | $77.40 \pm 0.06$ | $43.63 \pm 0.34$ | $42.69 \pm 0.03$ | $65.95 \pm 1.32$ | $63.73 \pm 0.98$ | $61.89 \pm 3.91$ |
| APPNP | $77.78 \pm 0.93$ | $67.42 \pm 1.31$ | $74.52 \pm 0.49$ | $42.15 \pm 0.17$ | $42.47 \pm 0.03$ | $68.38 \pm 4.37$ | $65.10 \pm 1.71$ | $64.59 \pm 3.30$ |
| JKNET | $78.20 \pm 0.20$ | $66.80 \pm 0.33$ | $75.62 \pm 0.37$ | $48.16 \pm 0.25$ | $42.21 \pm 0.05$ | $60.00 \pm 2.36$ | $42.55 \pm 2.92$ | $39.73 \pm 2.72$ |
| DAGNN | $65.98 \pm 1.49$ | $60.04 \pm 1.98$ | $72.39 \pm 0.90$ | $33.39 \pm 0.19$ | $40.61 \pm 0.03$ | $61.35 \pm 1.73$ | $57.45 \pm 1.97$ | $44.87 \pm 3.24$ |
| Feature-SBP | $82.46 \pm 0.07$ | $70.63 \pm 0.52$ | $77.41 \pm 0.21$ | $49.16 \pm 0.19$ | $42.31 \pm 0.03$ | $78.38 \pm 0.00$ | $80.39 \pm 0.00$ | $72.97 \pm 0.00$ |
| Label-SBP | $82.90 \pm 0.00$ | $73.04 \pm 0.10$ | $80.32 \pm 0.04$ | $45.60 \pm 0.11$ | $42.41 \pm 0.02$ | $78.38 \pm 0.00$ | $80.39 \pm 0.00$ | $70.27 \pm 0.00$ |

- **RQ1** How does SBP perform in node classification tasks using standard-depth GNNs?

- **RQ2** How effectively does SBP mitigate oversmoothing in deep layers?

- **RQ3** How sensitive, robust, and scalable is SBP to different hyperparameters, model backbones, and graph homophily levels?

**Datasets.** We use nine widely-used node classification benchmark datasets (Table 7), where four of them are heterophilic (Texas, Wisconsin, Cornell, Squirrel, and Amazon-rating [36]), and the remaining four are homophilic (Cora [37], Citeseer [38], and PubMed [39]), including one large-scale dataset (ogbn-arxiv [40]). Details of these datasets, including their homophily levels, are summarized in Table 1. We also experiment on the Contextual Stochastic Block Model (CSBM) [34] to show the performance of SBP on different homophily levels with detailed settings in Appendix I.1.

Table 1: Summary of datasets. $H(G)$ denotes the edge homophily level, with higher values indicating more homophilic graphs.

| Dataset | $H(G)$ | Classes | Nodes | Edges |
|---|---|---|---|---|
| **Cora** | 0.81 | 7 | 2,708 | 5,429 |
| **Citeseer** | 0.74 | 6 | 3,327 | 4,732 |
| **PubMed** | 0.80 | 3 | 19,717 | 44,338 |
| **Texas** | 0.21 | 5 | 183 | 295 |
| **Cornell** | 0.30 | 5 | 183 | 280 |
| **Amazon-ratings** | 0.38 | 5 | 24,492 | 93,050 |
| **Wisconsin** | 0.11 | 5 | 251 | 466 |
| **Squirrel** | 0.22 | 4 | 198,493 | 2,089 |
| **ogbn-arxiv** | 0.65 | 40 | 16,9343 | 1,166,243 |

**Baselines and Experiment Settings.** We implement SBP along with the following 10 baselines, all using the Simplified Graph Convolution Network (SGC) as the backbone GNN to ensure a fair comparison. 1) **Classic models**: MLP, vanilla SGC [9]. 2) **Normalization methods**: BatchNorm [41], PairNorm [17] and ContraNorm [16]. 3) **Edge dropping methods**: DropEdge [22]. 4) **Residual connections**: Residual, APPNP [42], JKNET [18] and DAGNN [19].

All methods are trained using the same setting, following [43]. For SBP, we select the optimal value of $\lambda$ from the set $\{0.1, 0.5, 0.9\}$, fix $\alpha = 1$, and then choose the best value for $\beta$ from $\{0.1, 0.5, 0.9\}$. Ablation studies on the influence of hyperparameters and the effectiveness of SBP with other GNN backbones can be found in Section 5.3.

## 5.1 RQ1: Node Classification Performance Using Standard-Depth GNNs

To evaluate the effectiveness of SBP under typical GNN settings, we assess its performance on node classification tasks using standard-depth models. Table 2 reports the mean node classification accuracy and standard deviation across 10 random seeds, using a 2-layer SGC backbone [44]. The results show that SBP improves node classification performance in standard-depth GNNs, yielding an average gain of 3 percentage points on homophilic graphs and 5 points on heterophilic graphs. Overall, SBP achieves superior performance across 8 datasets, with Label/Feature-SBP attaining the highest accuracy on 7 datasets.

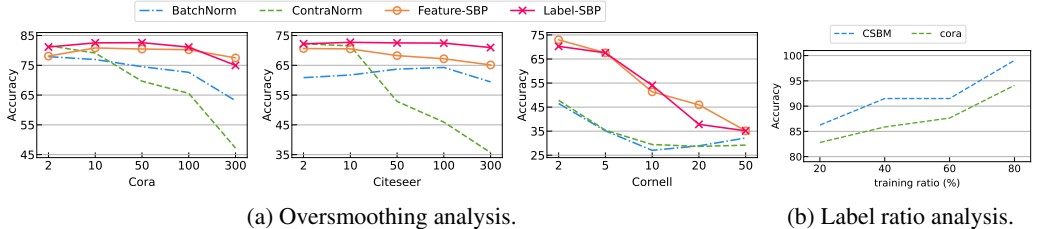

(a) Oversmoothing analysis.

(b) Label ratio analysis.

Figure 3: (a) Model performance under varying the number of layers. SBP remains effective up to 300 layers, while normalization methods degrade with depth due to oversmoothing. The X-axis denotes number of layers and the Y-axis denotes accuracy. (b) Sensitivity of Label-SBP to training label ratio. Label-SBP's performance improves with increasing label ratio, aligning with our theory. The X-axis denotes training label ratio and the Y-axis denotes accuracy.

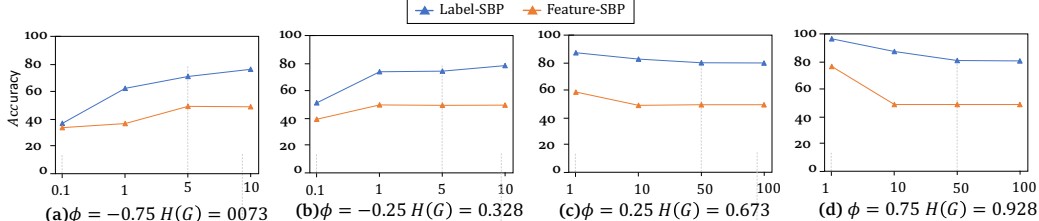

Figure 4: Impact of negative edge strength $\beta$ in SBP under different homophily levels on CSBM. $\phi$ controls the homophily level $H(G)$. The X-axis denotes $\beta$ and the Y-axis denotes accuracy. Homophilic graphs favor smaller $\beta$, while heterophilic graphs benefit from larger $\beta$ values.

## 5.2 RQ2: Anti-Oversmoothing Analysis

We further evaluate SBP's ability to counter oversmoothing in deep GNNs by varying the number of layers $K$. Figure 3(a) compares Label-/Feature-SBP against BatchNorm and PairNorm on both homophilic (Cora, Citeseer) and heterophilic (Cornell) benchmarks. For homophilic graphs, we test $K \in \{2, 10, 50, 100, 300\}$, and for the heterophilic graph, $K \in \{2, 5, 10, 20, 50\}$.

Notably, SBP maintains strong performance even at 300 layers, effectively mitigating oversmoothing in deep GNNs. In contrast, normalization-based methods exhibit substantial performance degradation as depth increases, reaffirming their vulnerability to oversmoothing [23]. Interestingly, we also observe that to maintain performance on heterophilic datasets, SBP requires a larger repulsion strength $\beta$ than typical settings (e.g., $\beta \in \{0.1, 0.5, 0.9\}$). As shown in Table 12, setting $\beta > 1$ enables SBP to sustain approximately 60% accuracy in deep-layer regimes on the Cornell dataset.

## 5.3 RQ3: Robustness, Sensitivity, and Scalability of SBP

**Sensitivity of Label-SBP to Training Label Ratio.** Since Label-SBP relies on ground-truth labels to construct the negative graph, we conduct an ablation study to examine its sensitivity to the proportion of labeled training data. As shown in Figure 3(b), Label-SBP's performance on the CSBM and Cora datasets improves with increasing label ratio when using a 2-layer SGC backbone. This aligns with our theoretical result that greater label availability leads to better structural balance in the signed graph, enhancing classification performance.

Nonetheless, even with a modest training ratio of 20%, Label-SBP achieves over 80% accuracy, while models trained with 80% labels approach 100% accuracy. Furthermore, as shown in Table 2, Label-SBP outperforms existing methods even under the default training splits of standard benchmarks, highlighting its robustness and practical effectiveness in real-world graph settings.

**Analysis of Negative Edge Strength $\beta$ Under Different Homophilic and Heterophilic Levels.** In order to evaluate the performance of SBP on graphs with arbitrary levels of homophily, we conduct an ablation study in the CSBM setting with controllable homophilic and heterophilic levels, following the setup from [31]. We examine a wide range of $\beta$ values, while keeping $\lambda = 0.5$, $\alpha = 1$ and using a hyperparameter in CSBM, $\phi$, to control the homophily level. The graph homophily is measured by $H(G)$.

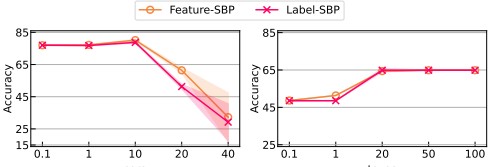

Figure 5: Impact of negative edge strength $\beta$ in SBP on real graphs. The X-axis denotes $\beta$ and the Y-axis denotes accuracy.

Table 3: Node classification accuracy (%) on the large-scale dataset *ogbn-arxiv*. Best results are highlighted in blue.

| Model | #L=2 | #L=4 | #L=8 | #L=16 |
|---|---|---|---|---|
| GCN | 67.32 ± 0.28 | 67.79 ± 0.25 | 65.54 ± 0.31 | 59.13 ± 0.95 |
| BatchNorm | 70.14 ± 0.28 | 70.93 ± 0.15 | 70.14 ± 0.43 | 63.24 ± 1.40 |
| PairNorm | 70.48 ± 0.20 | 71.59 ± 0.17 | 71.24 ± 0.07 | 68.92 ± 0.43 |
| ContraNorm | OOM | OOM | OOM | OOM |
| DropEdge | 64.07 ± 0.32 | 63.92 ± 0.27 | 60.74 ± 0.45 | 52.52 ± 0.34 |
| Residual | 66.90 ± 0.14 | 66.67 ± 0.25 | 61.76 ± 0.62 | 53.25 ± 0.75 |
| Label-SBP-v2 | 70.55 ± 0.22 | 71.54 ± 0.18 | 71.07 ± 0.28 | 69.33 ± 0.59 |

Table 4: Performance of SBP with different GNN backbones. Best results are highlighted in blue.

| | | #L=2 | #L=4 | #L=8 | #L=16 |
|---|---|---|---|---|---|
| Cora | GCN | 80.68 ±0.09 | 79.69 ±0.00 | 74.32 ±0.00 | 30.95 ±0.00 |
| | +Feature-SBP | 80.44 ±0.83 | 79.26 ±1.18 | 78.56 ±0.59 | 77.22 ±0.55 |
| | +Label-SBP | 80.31 ±0.70 | 79.16 ±1.30 | 79.50 ±0.00 | 77.43 ±1.49 |
| | GCNII | 78.58 ± 0.00 | 77.76 ± 0.24 | 73.47 ± 3.82 | 78.12 ± 1.32 |
| | +Label-SBP | 78.74 ± 1.54 | 78.87 ± 0.00 | 79.14 ± 0.35 | 79.17 ± 0.41 |
| | +Feature-SBP | 77.95 ± 0.91 | 78.82 ± 0.19 | 78.11 ± 1.62 | 78.82 ± 0.29 |
| Citesser | GCN | 67.45 ±0.54 | 65.62 ±0.25 | 37.22 ±2.46 | 22.03 ±4.76 |
| | +Feature-SBP | 67.38 ±0.66 | 66.94 ±0.00 | 66.29 ±0.02 | 65.35 ±1.99 |
| | +Label-SBP | 67.23 ±0.64 | 66.72 ±0.00 | 66.29 ±0.89 | 65.50 ±2.13 |
| | GCNII | 61.66 ± 0.67 | 63.23 ± 2.31 | 64.58 ± 2.66 | 66.21 ± 0.64 |
| | +Label-SBP | 65.31 ± 0.63 | 63.93 ± 3.66 | 68.33 ± 0.99 | 66.46 ± 0.00 |
| | +Feature-SBP | 65.63 ± 0.87 | 64.43 ± 3.55 | 68.44 ± 1.19 | 66.94 ± 0.00 |

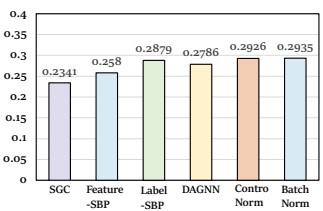

Figure 6: Real runtime (in seconds) of various methods.

Figure 4 shows the performance of Feature/Label-SBP across different $\beta$ values. In Figures 4(a) and (b), where $\phi < 0$ indicates heterophilic graphs, increasing $\beta$ significantly improves performance. Conversely, in Figures 4(c) and (d), where $\phi > 0$ corresponds to homophilic graphs, performance deteriorates as $\beta$ increases. We observe similar trends on real-world homophilic and heterophilic graph datasets, as shown in Figure 5. These results further highlight the role of $\beta$ as a repulsive force in the message-passing process, supporting our signed graph perspective for understanding and mitigating oversmoothing.

**Performance on Large-Scale Dataset.** To preserve graph sparsity and reduce computational overhead, we adopt SBP variants designed for large-scale graphs, as detailed in Section 4. Results on the ogbn-arxiv dataset are shown in Table 3. Overall, Label-SBP-v2 matches or outperforms existing normalization methods, particularly in the deep setting $L = 16$. These results demonstrate the empirical robustness and scalability of SBP-v2, which effectively leverages label information to mitigate oversmoothing even at large scale.

**SBP with Different GNN Backbones.** Beyond the SGC backbone, SBP can be seamlessly integrated into other GNN architectures, consistently yielding performance gains. Table 4 shows the results for SBP applied to GCN and GCNII across various model depths. SBP improves GCN performance by up to 47 points, particularly at depth $L = 16$, and also boosts the performance of GCNII, a state-of-the-art model explicitly designed to address oversmoothing. These results further support our insight that promoting structural balance is an effective strategy for mitigating oversmoothing in deep GNNs.

**Time Efficiency of SBP.** SBP is highly efficient, adding only minimal overhead compared to vanilla backbones. Figure 6 reports the actual runtime of various methods integrated with SGC on CSBM. Feature-SBP is the fastest among all methods, while Label-SBP ranks third—slightly slower than DAGNN but still more efficient than the normalization-based approaches. Additional analysis of SBP's runtime on large-scale graphs is provided in Appendix I.4.6.

## 6 Conclusion

In this work, we present a unified signed graph perspective on oversmoothing in GNNs, identifying structural balance as an ideal condition for preserving expressive node representations. Building on this insight, we propose Structural Balanced Propagation (SBP), a simple and plug-and-play method that constructs signed graphs to promote structural balance and mitigate oversmoothing. Extensive experiments demonstrate SBP's robustness, scalability, and consistent performance gains

across diverse settings. Beyond practical improvements, our work provides a theoretical foundation for understanding message-passing dynamics beyond vanilla GNNs and opens new directions for principled signed message-passing design.

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

# Appendix

## Contents

# A  Limitations

This work evaluates a wide range of GNNs. However, due to the diversity of GNN methods, it is impractical to assess all of them. Therefore, the proposed method focuses on classic techniques that utilize normalization, dropout, and residual connections. Moreover, due to limited computational resources and time, the proposed method has not been evaluated on super-large graphs, such as those with 1G nodes. SBP focuses on the oversmoothing problem, which leads to degraded performance in GNNs. Other factors that can negatively impact GNN performance are not discussed in this paper.

# B  Background on GNNs

## B.1  Graph Convolution Networks

To deal with non-Euclidean graph data, Graph Convolution Networks (GCNs) are proposed for direct convolution operation over graph, and have drawn interests from various domains. GCN is firstly introduced for a spectral perspective [8], but soon it becomes popular as a general message-passing algorithm in the spatial domain. In the feature transformation stage, GCN adopts a non-linear activation function (e.g., ReLU) and a layer-specific learnable weight matrix $W$ for transformation. The propagation rule of GCN can formulated as follow:

$$H_{(k)} = ReLU((\hat{A}H_{(k-1)})W_{(k)})  \tag{6}$$

## B.2  Simplified Graph Convolution Networks

Simplified Graph Convolution Networks (SGC [9]) simplifies and separates the two stages of GCNs: feature propagation and (non-linear) feature transformation. It finds that utilizing only a simple logistic regression after feature propagation (removing the non-linearities), which makes it a linear GCN, can obtain comparable performance to canonical GCNs. The propagation rule of GCN can formulated as follow:

$$H_{(k)} = \hat{A}H_{(k-1)})W_{(k)} = \hat{A}^k H_{(0)})W_{(k)}...W_{(1)}  \tag{7}$$

Moreover, SGC transforms $W_{(k)}...W_{(1)}$ to a general learnable parameter $W$, so the formula of SGC can be this:

$$H_{(k)} = \hat{A}^k H_{(0)})W  \tag{8}$$

## B.3  Signed Graph Propagation

Classical GNNs [8, 9, 10, 11] primarily focused on message-passing on unsigned graphs or graphs composed solely of positive edges. For example, if there exists a edge $\{i, j\}$ or the sign of edge $\{i, j\}$ is positive, the node $x_i$ updates its value by:

$$\hat{x}_i = x_i + \alpha(x_j - x_i) = (1 - \alpha)x_i + \alpha x_j, \alpha \in (0, 1).  \tag{9}$$

Compared to the unsigned graph, a signed graph extends the edges to either positive or negative. Notably, if the sign of edge $\{i, j\}$ is negative, the node $x_i$ update its value using the following expression:

$$\hat{x}_i = x_i - \beta(x_j - x_i) = (1 + \beta)x_i - \beta x_j, \beta \in (0, 1).  \tag{10}$$

In words, the positive interaction equation 9 indicates the attraction while the negative interaction equation 10 indicates that the nodes will repel their neighbors.

More generally, when considering all of the neighbors of node $x_i$, let $N_i^+$ denote the positive neighbor set while $N_i^-$ denote the negative neighbor set, where $N_i^+ \cup N_i^- = N_i$ and $N_i^+ \cap N_i^- = \emptyset$. The representation of $x_i$ is thus updated by:

$$\hat{x}_i = (1 - \alpha + \beta)x_i + \frac{\alpha}{|N_i^+|} \sum_{j \in N_i^+} x_j - \frac{\beta}{|N_i^-|} \sum_{j \in N_i^-} x_j .  \tag{11}$$

In particular, the two parameters $\alpha$ and $\beta$ mark the strength of positive and negative edges, respectively.

For further proofs of the theorems and propositions in the paper, we give a more simple and detailed definition in this section.

Let $D_{G^+} = \text{diag}(deg_1^+, \ldots, deg_n^+)$ and $D_{G^-} = \text{diag}(deg_1^-, \ldots, deg_n^-)$ be the degree matrices of the positive subgraph and negative subgraph, respectively. Let $A_{G^+}$ be the adjacency matrix of the graph $G^+$ with $[A_{G^+}]_{ij} = 1$ if $\{i, j\} \in E^+$ and $[A_{G^+}]_{ij} = 0$ otherwise. The adjacency matrix $A_{G^-}$ of the negative subgraph $G^-$ is defined by $[A_{G^-}]_{ij} = -1$ for $\{i, j\} \in E^-$ and $[A_{G^-}]_{ij} = 0$ for $\{i, j\} \notin E^-$.

The Laplacian plays a central role in the algebraic representation of structural properties of graphs. In the presence of negative edges, more than one definition of Laplacian is possible; see [25]. The Laplacian of the positive subgraph $G^+$ is $L_{G^+} := D_{G^+} - A_{G^+}$, while for the negative subgraph $G^-$ the following two variants can be used: $L_{G^-}^o := D_{G^-} - A_{G^-}$ and $L_{G^-}^r := -D_{G^-} - A_{G^-}$. Consequently, we have the following definitions.

Definition 1. Given the signed graph $G$, its opposing Laplacian is defined as
$$L_G^o := L_{G^+} + L_{G^-}^o = D_{G^+} + D_{G^-} - A_{G^+} - A_{G^-}, \tag{12}$$
and its repelling Laplacian is defined as
$$L_G^r = L_{G^+} + L_{G^-}^r = D_{G^+} - D_{G^-} - A_{G^+} - A_{G^-}. \tag{13}$$

Time is slotted at $t = 0, 1, \ldots$. Each node $i$ holds a state $x_i(t) \in \Omega$ at time $t$ and interacts with its neighbors at each time to revise its state. The interaction rule is specified by the sign of the links. Let $\alpha, \beta \geq 0$. We first focus on a particular link $\{i, j\} \in E$ and specify for the moment the dynamics along this link isolating all other interactions.

The DeGroot Rule:
$$x_s(t+1) = x_s(t) + \alpha(x_{-s}(t) - x_s(t)) = (1 - \alpha)x_s(t) + \alpha x_{-s}(t), \tag{14}$$
where $-s \in \{i, j\} \setminus \{s\}$ with $\alpha \in (0, 1)$

The Opposing Rule:
$$x_s(t+1) = x_s(t) + \beta(-x_{-s}(t) - x_s(t)) = (1 - \beta)x_s(t) - \beta x_{-s}(t); \tag{15}$$
or The Repelling Rule:
$$x_s(t+1) = x_s(t) - \beta(x_{-s}(t) - x_s(t)) = (1 + \beta)x_s(t) - \beta x_{-s}(t). \tag{16}$$

The Repelling Negative Dynamics:
$$\begin{aligned}
x_i(t+1) &= x_i(t) + \alpha \sum_{j \in N_i^+} (x_j(t) - x_i(t)) - \beta \sum_{j \in N_i^-} (x_j(t) - x_i(t)) \\
&= (1 - \alpha deg_i^+ + \beta deg_i^-)x_i(t) + \alpha \sum_{j \in N_i^+} x_j(t) - \beta \sum_{j \in N_i^-} x_j(t).
\end{aligned} \tag{17}$$

Denote $\mathbf{x}(t) = (x_1(t) \ldots x_n(t))^T$. We can now rewrite 17 in the compact form

$$\mathbf{x}(t+1) = M_G \mathbf{x}(t) = (I - \alpha L_{G_+} - \beta L_{G_-}^r)\mathbf{x}(t). \tag{18}$$

Here,
$$M_G = I - \alpha L_{G^+} - \beta L_{G^-}^r = I - L_G^{rw}, \tag{19}$$
with $L_G^{rw} = \alpha L_{G^+} + \beta L_{G^-}^r$ being the repelling weighted Laplacian of $G$. From Equation 18, $M_G \mathbf{1} = \mathbf{1}$ always holds. We present the following result, which by itself is merely a straightforward look into the spectrum of the repelling Laplacian $L_G^{rw}$.

## C    Proof of Theorem 3.2

Now consider the combined theorem.

**Theorem C.1** *Suppose that the positive edges are connected. Then along Equation 17 for any $0 < \alpha < 1/\max_{i \in V} \deg_i^+$, there exists a critical value $\beta_* \geq 0$ for $\beta$ such that*

*(i) if $\beta < \beta_*$, then we have $\lim_{t \to \infty} x_i(t) = \sum_{j=1}^n x_j(0)/n$ for all initial values $x(0)$;*

*(ii) if $\beta > \beta_*$, then $\lim_{t \to \infty} \|x(t)\| = \infty$ for almost all initial values w.r.t. Lebesgue measure.*

**Proof.** we change the signed graph update to the equivalent version of $x_i(t)$ read as:

$$x_i(t+1) = x_i(t) + \alpha \sum_{j \in N_i^+} (x_j(t) - x_i(t)) - \beta \sum_{j \in N_i^-} (x_j(t) - x_i(t)).$$

This can be expressed as:

$$x(t+1) = (1 - \alpha \deg^+ + \beta \deg^-)x_i(t) + \alpha \sum_{j \in N^+} x_j(t) - \beta \sum_{j \in N^-} x_j(t). \tag{20}$$

Algorithm 20 can be written as:

$$x(t+1) = M_G x(t) = (I - \alpha L_G^+ - \beta L_G^-)x(t). \tag{21}$$

Here, $M_G = I - \alpha L_G^+ - \beta L_G^-$, with $L_G^+ = \alpha L_C^+ + \beta L_C^-$ being the repelling weighted Laplacian of $G$, defined in Sec.B.3. From Eq.equation 21, $M_G \mathbf{1} = \mathbf{1}$ always holds. We present the following result, which by itself is merely a straightforward look into the spectrum of the repelling Laplacian $L_G^-$.

So theorem C.1 can be changed to the following version:

Suppose $G^+$ is connected. Then along Eq.equation 21 for any $0 < \alpha < 1/\max_{i \in V} \deg_i^+$, there exists a critical value $\beta > 0$ for $\beta$ such that:

 (i) if $\beta < \beta_*$, then average consensus is reached in the sense that $\lim_{t \to \infty} x_i(t) = \frac{1}{n} \sum_{j=1}^n x_j(0)$ for all initial values $x(0)$;

 (ii) if $\beta > \beta_*$, then $\lim_{t \to \infty} \|x(t)\| = \infty$ for almost all initial values w.r.t. Lebesgue measure.

**Proof.** Define $J = 11^T/n$. Fix $\alpha \in (0, 1/\max_{i \in V} \deg_i^+)$ and consider $f(\beta) = \lambda_{\max}(I - \alpha L_G^+ - \beta L_G^- - J)$, and $g(\beta) = \lambda_{\min}(I - \alpha L_G^+ - \beta L_G^- - J)$. The Courant–Fischer Theorem implies that both $f(\cdot)$ and $g(\cdot)$ are continuous and nondecreasing functions over $[0, \infty)$. The matrix $J$ always commutes with $I - \alpha L_G^+ - \beta L_G^-$, and 1 is the only nonzero eigenvalue of $J$. Moreover, the eigenvalue 1 of $J$ shares a common eigenvector 1 with the eigenvalue 1 of $I - \alpha L_G^+ - \beta L_G^-$.

Since $G^+$ is connected, the second smallest eigenvalue of $L_{G^+}$ is positive. Since $0 < \alpha < \frac{1}{\max_{i \in V} \deg_i^+}$, there holds $\lambda_{\min}(I - \alpha L_{G^+}) \geq -1$, again due to the Gershgorin Circle Theorem. Therefore, $f(0) < 1$, $g(0) \geq -1$. Noticing $f(\infty) = \infty > 1$, there exists $\beta_* > 0$ satisfying $f(\beta_*) = 1$. We can then verify the following facts:

 • There hold $f(\beta) < 1$ and $g(\beta) > -1$ if $\beta < \beta_*$. In this case, along Eq.equation 21 $\lim_{t \to \infty}(I - J)x(t) = 0$, which in turn implies that $x(t)$ converges to the eigenspace corresponding to the eigenvalue 1 of $M_G$. This leads to the average consensus statement in (i).

 • There holds $f(\beta) \geq 1$ if $\beta > \beta_*$. In this case, along Eq.equation 21 $x(t)$ diverges as long as the initial value $x(0)$ has a nonzero projection onto the eigenspace corresponding to $\lambda_{\max}(M_G)$ of $M_G$. This leads to the almost everywhere divergence statement in (ii).

The proof is now complete.

## D   Proof of Theorem 4.2

**Theorem D.1** *let $A > 0$ be a constant and define $\mathcal{F}(z)_c$ by $\mathcal{F}(z)_c = -c, z < -c$, $\mathcal{F}(z)_c = z, z \in [-c, c]$, and $\mathcal{F}(z)_c = c, z > c$. Define the function $\theta : E \to \mathbb{R}$ so that $\theta(\{i, j\}) = \alpha$ if $\{i, j\} \in E^+$ and $\theta(\{i, j\}) = -\beta$ if $\{i, j\} \in E^-$. Assume that node $i$ interacts with node $j$ at time $t$ and consider the following node interaction under the signed propagation rules:*

$$x_s(t+1) = \mathcal{F}(z)_c((1 - \theta)x_s(t) + \theta x_{-s}(t)), \ s \in \{i, j\}. \tag{22}$$

*let $\alpha \in (0, 1/2)$. Assume that $G$ is a structurally balanced complete graph under the partition $V = V_1 \cup V_2$. When $\beta$ is sufficiently large, for almost all initial values $x(0)$ w.r.t. Lebesgue measure, there exists a binary random variable $l(x(0))$ taking values in $\{-c, c\}$ such that*

$$\mathbb{P}\left(\lim_{t \to \infty} x_i(t) = l(x(0)), i \in V_1; \lim_{t \to \infty} x_i(t) = -l(x(0)), i \in V_2\right) = 1. \tag{23}$$

**Proof.** The proof is based on the following lemmas.

**Lemma D.2** *Fix $\alpha \in (0,1)$ with $\alpha \neq \frac{1}{2}$. For the dynamics 22 with the random pair selection process, there exists $\beta^*(\alpha) > 0$ such that*

$$\mathbb{P}\left(\limsup_{t \to \infty} \max_{i,j \in V} |x_i(t) - x_j(t)| = 2A\right) = 1$$

*for almost all initial beliefs if $\beta > \beta^*$.*

**Lemma D.3** *Fix $\alpha \in (1/2, 1)$ and $\beta \geq 2/(2\alpha - 1)$. Consider the dynamics 22 with the random pair selection process. Let $G$ be the complete graph with $\kappa(G^+) \geq 2$. Suppose for time $t$ there are $i_1, j_1 \in V$ with $x_{i_1}(t) = -c$ and $x_{j_1}(t) = c$. Then for any $\epsilon \in [0, (2\alpha - 1)c/2\alpha]$ and any $i_* \in V$, the following statements hold:*

(i) *There exist an integer $Z(\epsilon)$ and a sequence of node pair realizations, $G_{t+s}(\omega)$, for $s = 0, 1, \ldots, Z - 1$, under which $x_{i_*}(t + Z)(\omega) \leq -A + \epsilon$.*

(ii) *There exist an integer $Z(\epsilon)$ and a sequence of node pair realizations, $G_{t+s}(\omega)$, for $s = 0, 1, \ldots, Z - 1$, under which $x_{i_*}(t + Z)(\omega) \geq A - \epsilon$.*

**Proof.** From our standing assumption, the negative graph $G^-$ contains at least one edge. Let $k_*, m_* \in V$ share a negative link. We assume the two nodes $i_1, j_1 \in V$ labeled in the lemma are different from $k_*, m_*$, for ease of presentation. We can then analyze all possible sign patterns among the four nodes $i_1, j_1, k_*, m_*$. We present here just the analysis for the case with

$$\{i_1, k_*\} \in E^+, \{i_1, m_*\} \in E^+, \{j_1, k_*\} \in E^+, \{j_1, m_*\} \in E^+.$$

The other cases are indeed simpler and can be studied via similar techniques.

Without loss of generality we let $x_{m_*}(t) \geq x_{k_*}(t)$. First of all we select $G_t = \{i_1, k_*\}$ and $G_{t+1} = \{j_1, m_*\}$. It is then straightforward to verify that

$$x_{m_*}(t + 2) \geq x_{k_*}(t + 2) + 2\alpha c.$$

By selecting $G_{t+2} = \{m_*, k_*\}$ we know from $\beta \geq \frac{2}{(2\alpha - 1)} > \frac{1}{\alpha}$ that

$$x_{k_*}(t + 3) = -c, \quad x_{m_*}(t + 3) = c.$$

There will be two cases:

(a) Let $i_* \in \{m_*, k_*\}$. Noting that $\kappa(G^+) \geq 2$, there will be a path connecting to $k_*$ from $i_*$ without passing through $m_*$ in $G^+$. It is then obvious that we can select a finite number $Z_1$ of links which alternate between $\{m_*, k_*\}$ and the edges over that path so that $x_{i_*}(t+3+Z_1) \geq -c + \epsilon$. Here $Z_1$ depends only on $\alpha$ and $n$.

(b) Let $i_* \in \{m_*, k_*\}$. We only need to show that we can select pair realizations so that $x_{m_*}$ can get close to $-c$, and $x_{k_*}$ gets close to $c$ after $t + 3$. Since $G^+$ is connected, either $m_*$ or $k_*$ has at least one positive neighbor. For the moment assume $m'$ is a positive neighbor of $m_*$ and $k'$ is a positive neighbor of $k_*$ with $m' \neq k'$. Then from part (a) we can select $Z_2$ node pairs so that

$$x_{m_*}(t + 3 + Z_2) \leq -c + \epsilon, \quad x_{k_*}(t + 3 + Z_2) \geq c - \epsilon.$$

Thus, selecting the negative edge $\{m_*, k_*\}$ for $t+5+Z_2$ implies $x_{m_*}(t+6+Z_2) = c$ for $\beta \geq \frac{2}{(2\alpha-1)}$. The case with $m' = k'$ can be dealt with by a similar treatment, leading to the same conclusion.

This concludes the proof of the lemma.

In view of Lemmas D.2 and D.3, the desired theorem is a consequence of the second Borel–Cantelli Lemma.

# E  A Signed Graph Perspective on Existing Oversmoothing Countermeasures

We defined the signed graph propagation over the whole graph $\mathcal{G}$ written in the matrix form as:

$$\hat{X} = (1 - \alpha + \beta)X + \alpha\hat{A}^+X - \beta\hat{A}^-X, \tag{24}$$

Table 5: The mathematically equivalent raw normalized positive and negative adjacency matrices in signed graph propagation of various anti-oversmoothing methods.

| Method | Characteristic | Positive $\hat{A}^+$ | Negative $\hat{A}^-$ |
|---|---|---|---|
| GCN | $K$-layer graph convolutions | $\hat{A}$ | $0$ |
| SGC | $K$-layer linear graph convolutions | $\hat{A}$ | $0$ |
| BatchNorm | Normalized with column means and variance | $\hat{A}$ | $\mathbb{1}_n\mathbb{1}_n^T/n\hat{A}$ |
| PairNorm | Normalized with the overall means and variance | $\hat{A}$ | $\mathbb{1}_n\mathbb{1}_n^T/n\hat{A}$ |
| ContraNorm | Uniformed norm derived from contrastive loss | $\hat{A}$ | $(XX^T)\hat{A}$ |
| DropEdge | Randomized augmentation | $\hat{A}$ | $\hat{A}_m$ |
| Residual | Last layer connection | $\hat{A}$ | $I$ |
| APPNP | Initial layer connection | $\Sigma_{i=0}^{k+1}\alpha^i\hat{A}^i$ | $\alpha\Sigma_{j=0}^k\alpha^j\hat{A}^j$ |
| JKNET | Jumping to the last layer | $\Sigma_{i=0}^k\alpha^i\hat{A}^i + \hat{A}^{k+1}$ | $\Sigma_{j=0}^k\alpha^i\hat{A}^k$ |
| DAGNN | Adaptively incorporating different layer | $\Sigma_{i=0}^k\alpha^i\hat{A}^i + \hat{A}^{k+1}$ | $\Sigma_{j=0}^k\alpha^i\hat{A}^k$ |
| Feature-SBP (ours) | Label-induced negative graph | $\hat{A}$ | $\mathrm{softmax}(A_f^-)$ |
| Label-SBP (ours) | Feature-induced negative graph | $\hat{A}$ | $\mathrm{softmax}(A_l^-)$ |

where $\hat{A}^+$ is the raw normalized version of the positive adjacency matrix $A^+ \in \{0,1\}^{n\times n}$ and $\hat{A}^-$ is that of the negative adjacency matrix $A^- \in \{0,1\}^{n\times n}$.

We summarize eight specific methods with their corresponding positive and negative graphs in Table 5.

### E.1 Normalizations

**BatchNorm** BatchNorm centers the node representations $X$ to zero mean and unit variance and can be written as $\mathrm{BatchNorm}(x_i) = \frac{1}{\sqrt{\sigma^2+\epsilon}}(x_i - \frac{1}{n}\Sigma_{i=1}^n x_i)$, where $\epsilon > 0$ and $\sigma^2$ is the variance of node features. We rewrite BatchNorm in the signed graph propagation form as follows:

$$\hat{X} = \hat{A}X\Gamma_d^{-1} - \frac{\mathbb{1}_n\mathbb{1}_n^T}{n}\hat{A}X\Gamma_d^{-1} = \hat{A}\tilde{X} - \frac{\mathbb{1}_n\mathbb{1}_n^T}{n}\hat{A}\tilde{X}, \tag{25}$$

where $\Gamma_d = \mathrm{diag}(\sigma_1,\ldots,\sigma_d)$ is a diagonal matrix that represents column-wise variance with $\sigma_i^2 = \frac{1}{n}\sum_{j=1}^n ((\hat{A}X)_{ji} - \mathbb{1}_n^\top\hat{A}X/n)^2$, and $\tilde{X} = X\Gamma_d^{-1}$ is a normalized version of $X$. We can correspond to the positive graph $A^+$ to $\hat{A}$ and the negative graph $A^-$ to $\frac{\mathbb{1}_n\mathbb{1}_n^T}{n}\hat{A}$ in equation 25.

**PairNorm** We then introduce another method called PairNorm where the only difference between it and BatchNorm is that PairNorm scales all the entries in $X$ using the same number rather than scaling each column by its own variance. The formulation of PairNorm can be rewritten as follows:

$$\hat{X} = \frac{1}{\Gamma}\hat{A}X - \frac{1}{\Gamma}\frac{\mathbb{1}_n\mathbb{1}_n^T}{n}\hat{A}X = \frac{1}{\Gamma}(\hat{A}X - \frac{\mathbb{1}_n\mathbb{1}_n^T}{n}\hat{A}X), \tag{26}$$

where $\Gamma = \|(\hat{A} - \mathbb{1}_n\mathbb{1}_n^T/n)X\|_F/\sqrt{n}$. We observe that PairNorm shares the same positive and negative graphs (up to scale) as BatchNorm. Another normalization technique, ContraNorm, turns out to extend the negative graph to an adaptive one based on node feature similarities.

**Proposition E.1** *Consider the update:*

$$\hat{X} = AX - \frac{\mathbb{1}_n\mathbb{1}_n^T}{n}AX, \tag{27}$$

*where $A \in \{0,1\}^{n\times n}$ is the adjacency matrix. Define the overall signed graph adjacency matrix $A_s$ as $A - \frac{\mathbb{1}_n\mathbb{1}_n^T}{n}A$. Then we have that the signed graph is (weakly) structurally balanced only if the original graph can be divided into several isolated complete subgraphs.*

**Proof.** Assume that there is no isolated node and no node has edges with all the other nodes. $(A_s)_{i,j} = (A)_{i,j} - \frac{deg_j}{n}$. If $(A)_{i,j} = 1$, then we have $(A_s)_{i,j} > 0$. If $(A)_{i,j} = 0$, then we have $(A_s)_{i,j} < 0$.

If the nodes can be divided into several isolated complete subgraphs, then the nodes set $V = V_1 \cup V_2 \ldots V_m$, where $|V_i| > 1$, $m$ is the number of the isolated complete subgraphs. So only the nodes within the same set have edges, thus relative entries of $A_s > 0$, while nodes from different sets do not, thus relative entries of $A_s < 0$.

On the other hand, if $A_s$ is (weakly) structurally balanced, then the nodes set can be expressed as $V = V_1 \cup V_2 \ldots V_k$, where $|V_i| > 1$, $k$ is the number of the separated parties in the signed graph. The entry of $A_s$ in the same parties is positive, while between different parties is negative. According to $(A_s)_{i,j} = (A)_{i,j} - \frac{deg_j}{n}$, we know that nodes in the same parties are connected in the original graph while not connected in the original graph between different parties. So the graph can be divided into several isolated complete subgraphs.

Overall, the signed graph is (weakly) structurally balanced only if the original graph can be divided into several isolated complete subgraphs, the proof is over.

The Proposition shows that in order for the structural balance property to hold for the signed graph of normalization, the graph needs to satisfy an unrealistic condition where the edges strictly cluster the nodes.

**ContraNorm**   ContraNorm is inspired by the uniformity loss from contrastive learning, aiming to alleviate dimensional feature collapse. For simplicity, we consider the spectral version of ContraNorm that takes the following form:

$$\hat{X} = (1 + \alpha)\hat{A}X - \alpha/\tau(XX^T)\hat{A}X, \tag{28}$$

where $\alpha \in (0, 1)$ and $\tau > 0$ are hyperparameters. We can see that $\hat{A}$ is again the positive graph and $(XX^T)\hat{A}$ is the negative graph in the corresponding signed graph propagation.

Consider the update:

$$\hat{X} = AX - \frac{XX^T}{n}AX, \tag{29}$$

Define the overall signed graph adjacency matrix $A_s = A - \frac{XX^T}{n}A$ where $(A_s)_{i,j} = (A)_{i,j} - \frac{1}{n}\Sigma_{k=1}^n x_i x_k^T (A)_{k,j}$ .

Assume that the nodes feature is normalized every update, that is $||x_i||_2 = 1$ for every $i$.

If $(A)_{i,j} = 1$, then we have that

$$\begin{aligned}
(A_s)_{i,j} &= (A)_{i,j} - \frac{1}{n}\Sigma_{k=1}^n x_i x_k^T (A)_{k,j} \\
&= 1 - \frac{1}{n}\Sigma_{k=1}^n x_i x_k^T (A)_{k,j} \\
&> 1 - \frac{1}{n}\Sigma_{k=1}^n (A)_{k,j} \\
&= 1 - \frac{d_j}{n} > 0.
\end{aligned} \tag{30}$$

That means if $(A)_{i,j} = 1$, then $(A_s)_{i,j} > 0$. However, if $(A)_{i,j} = 0$, then we have that

$$\begin{aligned}
(A_s)_{i,j} &= (A)_{i,j} - \frac{1}{n}\Sigma_{k=1}^n x_i x_k^T (A)_{k,j} \\
&= -\frac{1}{n}\Sigma_{k=1}^n x_i x_k^T (A)_{k,j} \\
&= -\frac{1}{n}\Sigma_{k \in N_j} x_i x_k^T.
\end{aligned} \tag{31}$$

Intuitively, if $x_i$ has similar features to $x_j$'s neighbors, then we have that $(A_s)_{i,j} < 0$, which means trying to repel nodes with similar representations. If $x_i$ has different features to $x_j$'s neighbors,

then we have that $(A_s)_{i,j} > 0$, which means trying to aggregate nodes with original different representations.

If graph $G$ is a completed graph, then all entries of $(A_s) > 0$, however, when all of the nodes coverage to each other, $\Sigma_{k=1}^n x_i x_k^T (A)_{k,j} = \Sigma_{k=1}^n x_i x_k^T$ will also become bigger.

## E.2   Dropping

For DropMessage [21], it is a unified way of DropNode, DropEdge and Dropout but with a more flexible mask strategy. We have discussed the DropNode and DropEdge in our paper. DropMessage can be viewed as randomly dropping some dimension of the aggregated node features instead of the whole node or the whole edge. We give the unified positive and negative graph of DropMessage in the term of the signed graph. The propagation of DropMessage can be expressed as $H^{(k)} = AH^{(k-1)} - M_m$, where if dropping $AH_{ij}^{(k-1)}$, then $M_{ij} = AH_{ij}^{(k-1)}$ else $M_{ij} = 0$.

## E.3   Residual Connections

The standard residual connection [44, 45] directly combines the previous and the current layer features together. It can be formulated as:

$$\hat{X} = (1 - \alpha)X + \alpha\hat{A}X = X + \alpha\hat{A}X - \alpha IX. \tag{32}$$

For residual connections, the positive adjacency matrix is $\hat{A}$ and the negative adjacency matrix $I$ in the corresponding signed graph propagation.

**APPNP**   We reformulate the method APPNP [42] as the signed propagation form of the initial node feature. Another propagation process is APPNP [42] which can be viewed as a layer-wise graph convolution with a residual connection to the initial transformed feature matrix $X^{(0)}$, expressed as:

$$\hat{X}^{(k+1)} = (1 - \alpha)X^{(0)} + \alpha\hat{A}X^{(k)}. \tag{33}$$

**Theorem E.2** *With $\hat{A}^+ = \Sigma_{i=0}^{k+1}\alpha^i\hat{A}^i$ and $\hat{A}^- = \alpha\Sigma_{j=0}^k\alpha^j\hat{A}^j$, the propagation process of APPNP following the signed graph propagation.*

**Proof.** Easily prove with mathematical induction.

In addition to combining with the last and initial layer features, the last type integrates several intermediate layer features. The established representations are JKNET [18] and DAGNN [19].

**JKNET**   JKNET is a deep graph neural network which exploits information from neighborhoods of differing locality. JKNET selectively combines aggregations from different layers with Concatenation/Max-pooling/Attention at the output, i.e., the representations "jump" to the last layer. Using attention mechanism for combination at the last layer, the $k+1$-layer propagation result of JKNET can be written as:

$$\begin{aligned} X^{(k+1)} &= \alpha_0 X^{(0)} + \alpha_1 X^{(1)} + \cdots \alpha_k X^{(k)} \\ &= \Sigma_{i=0}^k \alpha_i \hat{A}^i X^{(0)}, \end{aligned} \tag{34}$$

where $\alpha_0, \alpha_1, \cdots, \alpha_k$ are the learnable fusion weights with $\Sigma_{i=0}^k\alpha_i = 1$.

**DAGNN**   Deep Adaptive Graph Neural Networks (DAGNN) [19] tries to adaptively add all the features from the previous layer to the current layer features with the additional learnable coefficients. After decoupling representation transformation and propagation, the propagation mechanism of DAGNN is similar to that of JKNET.

$$X^{(k+1)} = \Sigma_{i=0}^k \alpha_i \hat{A}^i H^{(0)}, \ H^{(0)} = f_\theta(X^{(0)}) \tag{35}$$

$H^{(0)} = f_\theta(X^{(0)})$ ) is the non-linear feature transformation using an MLP network, which is conducted before the propagation process and $\alpha_0, \alpha_1, \cdots, \alpha_k$ are the learnable fusion weights with $\Sigma_{i=0}^k\alpha_i = 1$.

**Theorem E.3** *With* $\hat{A}^+ = \Sigma_{i=0}^{k-1}\alpha^i\hat{A}^i + \hat{A}^k$ *and* $\hat{A}^- = \Sigma_{j=0}^{k-1}\alpha^j\hat{A}^k$, *the propagation process of JKNET and DAGNN following the signed graph propagation.*

**Proof.** Easily prove with mathematical induction.

As for more residual inspired methods [20, 46, 47, 48], we select GCNII and wGCN to give a detailed discussion as follows.

- As for GCNII [20], it is an improved version of APPNP with the learnable coefficients $\alpha_i$ and changes the learnable weight W to $(1-\beta_i)I + \beta_i W$ each layer, so it shares the same positive and negative graph as APPNP.

- As for the wGCN [46], it incorporates trainable channel-wise weighting factors $\omega$ to learn and mix multiple smoothing and sharpening propagation operators at each layer, same as the init residual combines but change parameters $\alpha$ to be learnable with a more detailed selection strategy.

# F  Oversmoothing Metrics

There exist a variety of different approaches to quantify over-smoothing in deep GNNs, here we choose the measure based on the Dirichlet energy on graphs [15, 49].

$$\epsilon(X(t)) = \frac{1}{v}\Sigma_{i\in V}\Sigma_{j\in N_i}||x_i(t) - x_j(t)||_2^2, \tag{36}$$

where $v$ is the number of the nodes, $x_i(t)$ is the node feature of node $i$ at time $t$. $N_i$ represents the neighbor set of node $i$, In the signed graph, it including nodes connected to $i$ by both positive and negative edges. Oversmoothing means that when the layers are infinity, all of the node features will converge, that is to say $\lim_{t\to\infty}\epsilon(X(t)) \to 0$.

In Theorem 3.2, there are 2 cases:

- $if \beta < \beta_*,$ then we have $\lim_{t\to\infty} x_i(t) = \sum_{j=1}^n x_j(0)/n$ for all initial values $x(0)$

- $if \beta > \beta_*,$ then $\lim_{t\to\infty} ||x(t)|| = \infty$ for almost all initial values w.r.t. Lebesgue measure.

In the first case, all the node features will coverage to the mean of them and therefore $\lim_{t\to\infty}\epsilon(X(t)) \to 0$, then oversmoothing happens. In the second case, the node features will diverge to infinity and thus $\lim_{t\to\infty}\epsilon(X(t)) \to 0$ or $\infty$ which is also not what we want.

Theorem 3.2 demonstrated that both insufficient repulsion and excessive repulsion caused by the negative graph can hinder performance in signed graph propagation. From this, we conclude that relying solely on the negative signs is insufficient to alleviate oversmoothing. Therefore, we propose the provable solution: a structurally balanced graph to efficiently alleviate oversmoothing in Theorem 4.2. Specifically, we have the following conclusion from the structurally balanced graph in Theorem 4.2:

$$\mathbb{P}\left(\lim_{t\to\infty} x_i(t) = l(x(0)), i \in V_1; \lim_{t\to\infty} x_i(t) = -l(x(0)), i \in V_2\right) = 1. \tag{37}$$

Then we have:

$$\lim_{t\to\infty}\epsilon(X(t)) = \lim_{t\to\infty}\frac{1}{v}\Sigma_{i\in V}\Sigma_{j\in N_i}||x_i(t) - x_j(t)||_2^2 \tag{38}$$

$$= \lim_{t\to\infty}\frac{1}{v}\Sigma_{i\in V_1}\Sigma_{j\in N_i}||x_i(t) - x_j(t)||_2^2 + \frac{1}{v}\Sigma_{i\in V_2}\Sigma_{j\in N_i}||x_i(t) - x_j(t)||_2^2 \tag{39}$$

$$= \lim_{t\to\infty}\frac{1}{v}\Sigma_{i\in V_1}\Sigma_{j\in N_i, y_i\neq y_j}||x_i(t) - x_j(t)||_2^2 + \frac{1}{v}\Sigma_{i\in V_2}\Sigma_{j\in N_i, y_i\neq y_j}||x_i(t) - x_j(t)||_2^2 \tag{40}$$

$$= \lim_{t\to\infty}\frac{1}{v}\Sigma_{i\in V_1}\frac{v}{2}\times 2c + \frac{1}{v}\Sigma_{i\in V_2}\frac{v}{2}\times 2c \tag{41}$$

$$= \lim_{t\to\infty}\frac{1}{v}(\frac{v}{2}\times\frac{v}{2}\times 2c + \frac{v}{2}\times\frac{v}{2}\times 2c) \tag{42}$$

$$= vc \geq 0 \tag{43}$$

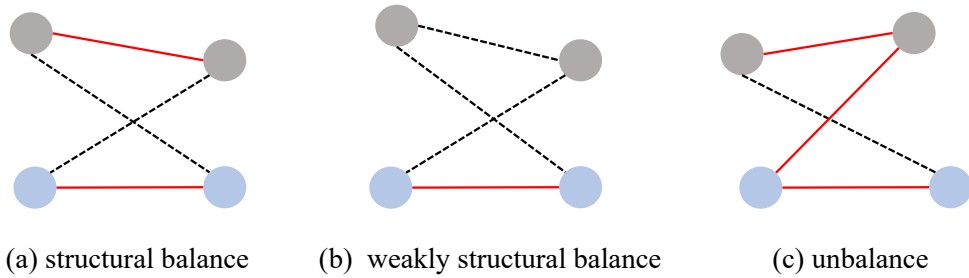

(a) structural balance    (b) weakly structural balance    (c) unbalance

Figure 7: Examples of structural balanced (left), weakly structural balanced (middle), and unbalanced signed graphs (right). Here red lines represent positive edges; black dashed lines represent negative edges; gray and blue circles represent nodes from different labels

So Theorem 4.2 proves that under certain conditions, structural balance can alleviate oversmoothing even when the layers are infinity.

## G   Weakly Structural Balance

To clarify the concept of structural balance, weakly structural balance and unbalance signed graph, we give the examples as shown in Figure 7. The notion of structural balance can be weakened in the following definition G.1.

**Definition G.1** *A signed graph $G$ is **weakly structurally balanced** if there is a partition of $V$ into $V = V_1 \cup V_2 \cup \ldots \cup V_m$, $m \geq 2$ with $V_1, \ldots, V_m$ being nonempty and mutually disjoint, where any edge between different $V_i$'s is negative, and any edge within each $V_i$ is positive.*

Then we show that when $\mathcal{G}$ is a complete graph, weak structural balance also leads to clustering of node states.

**Theorem G.2 ([25], Theorem 10)** *Assume that node $i$ interacts with node $j$ and $x_i(t)$ represents the value of node $i$ at time t. Let $\theta = \alpha$ if the edge $\{i, j\}$ is positive and $\theta = \beta$ if the edge $\{i, j\}$ is negative. Consider the constrained signed propagation update:*

$$x_i(t+1) = \mathcal{F}_c((1 - \theta)x_i(t) + \theta x_J(t)). \tag{44}$$

*Let $\alpha \in (0, 1/2)$. Assume that $\mathcal{G}$ is a weakly structurally balanced complete graph under the partition $V = V_1 \cup V_2 \cdots \cup V_m$. When $\beta$ is sufficiently large, for almost all initial values $x(0)$ w.r.t. Lebesgue measure, there exists $m$ random variable $l_1(x(0))$, $l_2(x(0))$, $\ldots$, $l_m(x(0))$, each of which taking values in $\{-c, c\}$ such that*

$$\mathbb{P}\left( \lim_{t \to \infty} x_i(t) = l_j(x(0)), i \in V_j, j = 1, \ldots, m \right) = 1. \tag{45}$$

## H   Statement and Proof of Label-SBP

In this section, we show that our method Label-SBP can create a structurally balanced graph under certain conditions and thus provably alleviate oversmoothing as the number of propagation steps increases. To achieve this, we introduce a metric, *structural imbalance degree* ($\mathcal{SID}$), to quantify the level of structural balance in arbitrary signed graph. Specifically, $\mathcal{SID}$ counts the number of edges that must be changed to achieve the structural balance.

**Definition H.1 (Structural Imbalance Degree)** *For each node $v$ in a signed graph $\mathcal{G}_s$ of $n$ nodes, let $\mathcal{P}(v)$ denote the subset of nodes that has the same label as $v$ but connected to $v$ by a non-positive edge; let $\mathcal{N}(v)$ denote the subset of nodes that has a different label from $v$ but connected*

Table 6: $\mathcal{SID}$ on CSBM (Contextual Stochastic Block Model ) with different methods. We set the two class means $u_1 = -1$ and $u_2 = 1$ respectively, the number of nodes $N = 100$, intra-class edge probability $p = 2\log 100/100$ and inter-class edge probability $q = \log 100/100$.

| Method | $\mathcal{P}_\downarrow$ | $\mathcal{N}_\downarrow$ | $\mathcal{SID}_\downarrow$ |
|---|---|---|---|
| DropEdge | 92.62 | 100.00 | 96.31 |
| Residual | 90.87 | 100.00 | 95.44 |
| GCN/SGC | 89.87 | 100.00 | 94.94 |
| APPNP | 0.00 | 100.00 | 50.00 |
| JKNET | 0.00 | 100.00 | 50.00 |
| DAGNN | 0.00 | 100.00 | 50.00 |
| BatchNorm | 89.87 | 4.56 | 47.22 |
| PairNorm | 89.87 | 4.56 | 47.22 |
| ContraNorm | 89.87 | 4.56 | 47.22 |
| Feature-SBP (ours) | 89.87 | 4.56 | 47.22 |
| Label-SBP (ours) | 32.46 | 36.16 | 34.31 |

*to $v$ by a non-negative edge. Then the structural imbalance degree of $\mathcal{G}$ is defined as $\mathcal{SID}(\mathcal{G}_s) = \frac{1}{2n}\sum_{v\in\mathcal{G}_s}(|\mathcal{P}(v)| + |\mathcal{N}(v)|)$.*

$\mathcal{SID}$ exhibits a fundamental characteristic: it increases as more edge signs deviate from the criteria of a structurally balanced graph, suggesting a higher degree of structural imbalance. Specifically, when the signed graph achieves the structural balance, we can assert that $\mathcal{SID} = 0$ as follows:

**Proposition H.2** *For a structural balanced complete graph $\mathcal{G}_{sb}$, we have $\mathcal{SID}(\mathcal{G}_{sb}) = 0$.*

**Proof** If the graph is structurally balanced, we can see that for a node $v$, $\mathcal{P}(v) = 0$ and $\mathcal{N}(v) = 0$ due to the structural balance complete graph assumption. So $\mathcal{SID}(\mathcal{G}) = 0$.

Based on the $\mathcal{SID}$, we can quantity the degree of structural balance in the equivalent signed graphs induced by anti-oversmoothing methods discussed in the previous section, as shown in Table 6. Our results show that previous anti-oversmoothing methods either remain a high $\mathcal{SID}$ or an imbalance $\mathcal{P}$ and $\mathcal{N}$. In contrast, our methods effectively reduce the $\mathcal{SID}$, resulting in a more structurally balanced graph, or at least be on par with previous methods.

Besides the empirical observation, we present the following theoretical result which demonstrates that Label-SBP can be guaranteed to achieve a certain level of structure balance:

**Proposition H.3** *Assuming balanced node label classes with $|Y_1| = |Y_2|$, a labeled node ratio denoted as $p$, and the signed graph $\mathcal{G}_s^l$ created by Label-SBP, then we have $\mathcal{SID}(\mathcal{G}_s^l) \leq (1-p)n/2$.*

**Proof** Without loss of generality, assume that the node feature has been normalized which means that $||x_i||_2 = 1$ for every $i$. If $x_i$ and $x_j$ has the same label, then we have that, $(A_s)_{i,j} = (A)_{i,j} + 1 > 1$. If $x_i$ and $x_j$ has different labels, then we have that $(A_s)_{i,j} = (A)_{i,j} - 1 \leq 0$.

We first prove that $\mathcal{SID}(\mathcal{G}, p) \leq (1-p)\frac{n}{2}$ where $n$ is the nodes number and $p$ is the label ratio. We have that

$$\mathcal{P}(v) + \mathcal{N}(v) \leq (1-p)n, \tag{46}$$

because for a single node $v$ only the remaining $(1-p)n$ nodes' labels are unknown and therefore their edges may need to change so that

$$\begin{aligned}
\mathcal{SID}(\mathcal{G}) &= \frac{1}{2n}\sum_{v\in\mathcal{G}}(|\mathcal{P}(v)| + |\mathcal{N}(v)|) \\
&\leq \frac{1}{2n}\sum_{v\in\mathcal{G}}(1-p)n \\
&= (1-p)\frac{n}{2}.
\end{aligned} \tag{47}$$

We know that when $\mathcal{SID}(\mathcal{G}) = 0$, then we have that the nodes $V$ set can be divided into $V_1 \cup V_1 \cdots \cup V_L$ where $L$ is the number of the node classes. There are only positive edges with the node subset and only negative edges between the node subset.

Since $C = 2$, the node set can be divided into $V_1$ and $V_2$, the signed graph is structurally balanced. According to Theorem 4.2, we have that the nodes in $V_1$ will converge to the $c$ where $||c||_2 = 1$ and the nodes in $V_2$ will converge to $-c$. Thus under Label-SBP propagation, the oversmoothing will only happen within the same label and repel different labels to the boundary.

Proposition H.3 suggests that Label-SBP constrains $\mathcal{SID}$ linearly with the training ratio $p$, indicating that $\mathcal{SID}$ diminishes with an increase in the labeling ratio $p$. In particular, it implies that Label-SBP can strictly establish a structurally balanced graph for any graph under the full supervision condition, making the model easier to distinguish nodes with different labels as the number of layers increases:

**Theorem H.4** *Under full supervision ($p = 1$), the signed graph $\hat{\mathcal{G}}_s^l$ induced by Label-SBP achieves $\mathcal{SID}(\hat{\mathcal{G}}_s^l) = 0$. Consequently, under the constrained signed propagation as given by equation 2, nodes from distinct classes will converge towards unique constants.*

$$\mathbb{P}\left(\lim_{k \to \infty} X_i^{(k)} = c, i \in \tilde{V}_1; \lim_{k \to \infty} X_i^{(k)} = -c, i \in \tilde{V}_2\right) = 1. \tag{48}$$

# I  Details of Experiments

## I.1  Details of the Dataset

Table 7: Summary of datasets. $H(G)$ refers to the edge homophily level: the higher, the more homophilic the dataset is.

| Dataset | $H(G)$ | Classes | Nodes | Edges |
|---|---|---|---|---|
| **Cora** | 0.81 | 7 | 2,708 | 5,429 |
| **Citeseer** | 0.74 | 6 | 3,327 | 4,732 |
| **PubMed** | 0.80 | 3 | 19,717 | 44,338 |
| **Texas** | 0.21 | 5 | 183 | 295 |
| **Cornell** | 0.30 | 5 | 183 | 280 |
| **Amazon-ratings** | 0.38 | 5 | 24,492 | 93,050 |
| **Wisconsin** | 0.11 | 5 | 251 | 466 |
| **Squirrel** | 0.22 | 4 | 198,493 | 2,089 |
| **Ogbn-Arxiv** | 0.65 | 40 | 16,9343 | 1,166,243 |

We consider two types of datasets: Homophilic and Heterophilic. They are differentiated by the *homophily level* of a graph.

$$\mathcal{H} = \frac{1}{|V|} \sum_{v \in V} \frac{\text{Number of } v\text{'s neighbors who have the same label as } v}{\text{Number of } v\text{'s neighbors}}.$$

The low homophily level means that the dataset is more heterophilic when most of the neighbors are not in the same class, and the high homophily level indicates that the dataset is close to homophilic when similar nodes tend to be connected. In the experiments, we use four homophilic datasets, including Cora, CiteSeer, PubMed, and Ogbn-Arxiv, and four heterophilic datasets, including Texas, Wisconsin, Cornell, Squirrel, and Amazon-rating [36]). The datasets we used covers various homophily levels.

**CSBM Settings.**    To quantify the structural balance of the mentioned methods, we simplified the graph to 2-CSBM($N, p, q, \mu_1, \mu_2, \sigma^2$) following [24]. It consists of two classes $\mathcal{C}_1$ and $\mathcal{C}_2$ of nodes of equal size, in total with $N$ nodes. For any two nodes in the graph, if they are from the same class, they are connected by an edge independently with probability $p$, or if they are from different classes, the probability is $q$. For each node $v \in \mathcal{C}_i, i \in \{1, -1\}$, the initial feature $X_v$ is sampled independently from a Gaussian distribution $\mathcal{N}(\mu_i, \sigma^2)$, where $\mu_i = \mathcal{C}_i, \sigma = I$. In this paper, we assign $N = 100$ and the feature dimension is 8.

## I.2 Experiments Setup

For the SGC backbone, we follow the [44] setting where we run 10 runs for the fixed seed $42$ and calculate the mean and the standard deviation. Furthermore, we fix the learning rate and weight decay in the same dataset and run 100 epochs for every dataset. For the GCN backbone, we follow the [16] settings where we run 5 runs from the seed $\{0, 1, 2, 3, 4\}$ and calculate the mean and the standard deviation. We fix the hidden dimension to $32$ and dropout rate to $0.6$. Furthermore, we fix the learning rate to be $0.005$ and weight decay to be $5e - 4$ and run 200 epochs for every dataset. We use the default splits in torch_geometric. We use Tesla-V100-SXM2-32GB in all experiments.

## I.3 Time Complexity Analysis and the Modified SBP

**Label-**SBP    As shown in equation 4.2, we maintain the positive adjacency matrix $A^+ = \hat{A}$ and construct the negative adjacency matrix $A_l$ by assigning 1 when nodes $i, j$ have different labels, -1 when they share the same label, and 0 when either label is unknown. We then apply softmax to $A_l$ to normalize the negative adjacency matrix. The overall signed adjacency matrix is $A_{sign} = \alpha A^+ - \beta softmax(A_l)$, where $\alpha$ and $\beta$ are hyperparameters. Given $n_t$ training nodes and $d$ edges in the graph, our Label-SBP increases the edge count from $O(d)$ to $O(n_t^2)$, thereby increasing the computational complexity to $O(n_t^2 d)$.

**Feature-**SBP    When labels are unavailable, we propose Feature-SBP, which uses the similarity matrix of node features to create the negative adjacency matrix. As depicted in equation 4.2, we design the negative adjacency matrix as $A_f = -X_0 X_0^T$. We then apply softmax to $A_f$ to normalize it. The overall matrix follows the same format as Label-SBP: $A_{sign} = \alpha A^+ - \beta softmax(A_f)$, where $\alpha$ and $\beta$ are hyperparameters. The additional computational complexity primarily stems from the negative graph propagation, which involves $X_0 X_0^T \in \mathbb{R}^{n \times n}$, increasing the overall complexity to $O(n^2 d)$.

We show the computation time of different methods in the Table 8. On average, we improve performance on 8 out of 9 datasets (as shown in Table 2) with less than 0.05s overhead—even faster than three other baselines. We believe this time overhead is acceptable given the benefits it provides.

Table 8: Estimated training time of SBP on Cora dataset. All experiments are run under 2 layers. s is the abbreviation for second. Precompute time is the aggregation time across layers, train time is the update time of the SGC weight $W$, total time is the sum of them.

|  | Label-SBP | Feature-SBP | BatchNorm | ContraNorm | Residual | JKNET | DAGNN | SGC |
|---|---|---|---|---|---|---|---|---|
| Precompute time | 0.1809s | 0.1520s | 0.1860s | 0.1888s | 0.0604s | 0.0577s | 0.1438s | 0.1307s |
| Train time | 0.1071s | 0.1060s | 0.1076s | 0.1038s | 0.1368s | 0.1446s | 0.1348s | 0.1034s |
| Total time | 0.2879s | 0.2580s | 0.2935s | 0.2926s | 0.1972s | 0.2023s | 0.2786s | 0.2341s |
| Rank | 6 | 4 | 8 | 7 | 1 | 2 | 5 | 3 |

**Scalability of** SBP **on large-scale graph**    For large-scale graphs, we introduce a modified version Label-SBP-v2 by only removing edges when pairs of nodes belong to different classes. This approach allows Label-SBP-v2 to eliminate the computational overhead of the negative graph, further enhancing the sparsity of large-scale graphs. For Feature-SBP, as the number of nodes $n$ increases, the complexity of this matrix operation grows quadratically, i.e., $\mathcal{O}(n^2 d)$. To address this, we reorder the matrix multiplication from $-X_0 X_0^T \in \mathbb{R}^{n \times n}$ to $-X_0^T X_0 \in \mathbb{R}^{d \times d}$. This preserves the distinctiveness of node representations across the feature dimension, rather than across the node dimension as in the original node-level repulsion. The modified version of Feature-SBP can be expressed as:

$$\text{(Feature-SBP-v2)} \quad X^k = (1 - \lambda)X^{(k-1)} + \lambda(\alpha \hat{A} X^{(K)} - \beta X^{(K)} \text{softmax}(-X_0^T X_0)) \quad (49)$$

This transposed alternative has a linear complexity in the number of samples, i.e., $\mathcal{O}(nd^2)$, significantly reducing the computational burden in cases where $n \gg d$.

We compare the compute time SBP with other baselines on ogbn-arxiv dataset over 100 epochs for a fair comparison. Among all the training times of the baselines, our Label-SBP-v2 achieves the 3rd fastest time while Feature-SBP-v2 ranks 5th. Therefore, we recommend using Label-SBP-v2 for large-scale graphs since they typically have a sufficient number of node labels. We believe that although there is a slight time increase, it is acceptable given the benefits.

Table 9: Estimated training time of SBP on ogbn-arixv dataset. All experiments are run under 2 layers and 100 epochs. s is the abbreviation for second.

| | Label-SBP | Feature-SBP | BatchNorm | ContraNorm | DropEdge | SGC |
|---|---|---|---|---|---|---|
| Train time (s) | 5.5850 | 6.1333 | 5.3872 | 5.8375 | 9.5727 | 5.3097 |
| Rank | 3 | 5 | 2 | 4 | 6 | 1 |

The code for the experiments will be available when our paper is acceptable. We will replace this anonymous link with a non-anonymous GitHub link after the acceptance. We implement all experiments in Python 3.9 with PyTorch Geometric on one NVIDIA Tesla V100 GPU.

## I.4 Results Analysis

### I.4.1 CSBM results

The comparative results of Label-SBP and Feature-SBP against SGC are presented in Table 10. As the number of layers increases, SGC's node features suffer from oversmoothing, causing the two classes to converge and accuracy to drop by nearly 30 points from its peak at 2 layers, down to $45\%$. Conversely, after 300 layers, SBP maintains strong performance, with node features of different classes repelling each other. This effect limits oversmoothing to within-class interactions, and improves performance from 85 to 91 in Label-SBP and from 48 to 82 in Feature-SBP, further substantiating our approach to mitigating oversmoothing.

We visualize the node features learned by Label-SBP in Figure 9. We can see that from layer 0 to layer 200, the node features from different labels repel each other and aggregate the node features from the same labels. And we also visualize the adjacency matrix of Label-SBP and Feature-SBP in Figure 10 and Figure 11 respectively, further verifying the effectiveness of our theorem and insights.

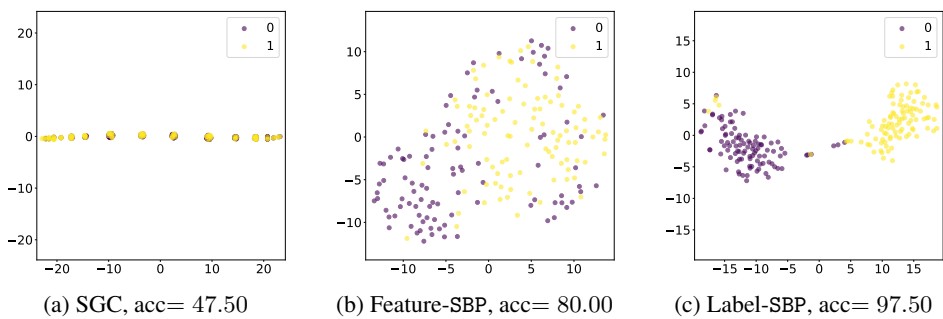

(a) SGC, acc= $47.50$     (b) Feature-SBP, acc= $80.00$     (c) Label-SBP, acc= $97.50$

Figure 8: The t-SNE visualization of the node features and the classification accuracy from 2-CSBM and Layer= 300. Left is the result of the vallina SGC, and the middle and right are the results of SBP.

Table 10: CSBM test accuracy (%) comparison results. The best results are marked in blue on each layer. The second best results are marked in gray on each layer. We run 10 runs for the seed from $0 - 9$ and demonstrate the mean $\pm$ std in the table.

| Model | #L=2 | #L=5 | #L=10 | #L=20 | #L=50 | #L=100 | #L=200 |
|---|---|---|---|---|---|---|---|
| SGC | $73.25 \pm 6.90$ | $44.50 \pm 9.34$ | $45.75 \pm 9.36$ | $45.75 \pm 9.36$ | $45.75 \pm 9.36$ | $45.75 \pm 9.36$ | $45.75 \pm 9.36$ |
| Feature-SBP | $48.75 \pm 5.62$ | $53.75 \pm 6.45$ | $63.75 \pm 6.25$ | $77.00 \pm 5.45$ | $82.00 \pm 4.58$ | $82.50 \pm 5.12$ | $82.00 \pm 5.45$ |
| Label-SBP | $85.75 \pm 4.04$ | $93.50 \pm 4.06$ | $93.50 \pm 3.57$ | $93.50 \pm 3.57$ | $92.25 \pm 3.44$ | $93.25 \pm 3.72$ | $91.25 \pm 6.05$ |

### I.4.2 GCN Results

The results for GCN are detailed in Table 11, respectively. Overall, SBP consistently outperforms all previous methods, especially in deeper layers. Beyond 16 layers in GCN, SBP maintains superior performance, affirming the effectiveness of our approach. Notably, SBP exceeds the best results of prior methods by at least $10\%$ and up to $30\%$ points in GCN's deepest layers, marking significant improvements. Moreover, unlike previous methods that perform best in shallow layers, SBP excels in

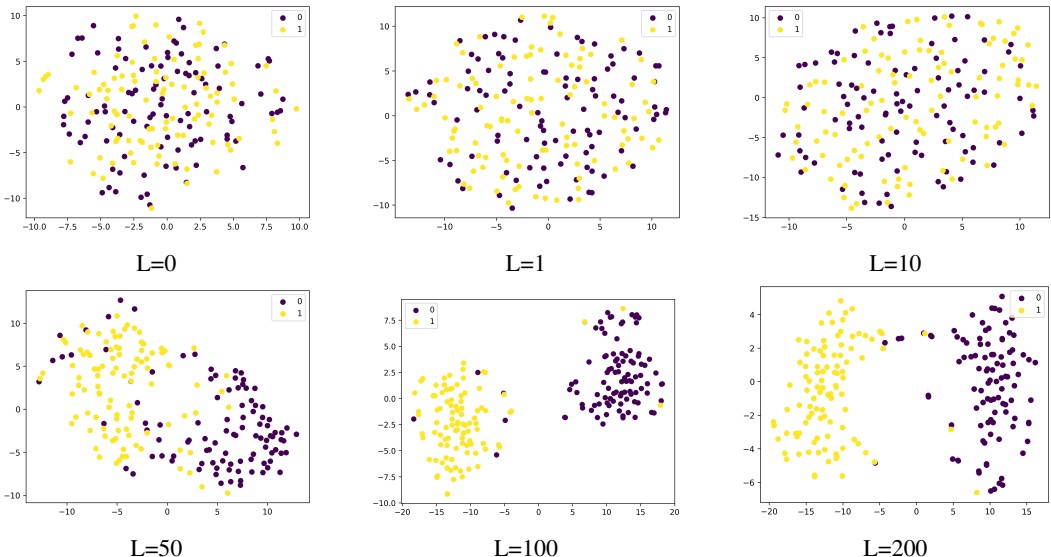

Figure 9: CSBM node features visualization. We update the features by Label-SBP. L is the propagation layer number. 0,1 represent different classes.

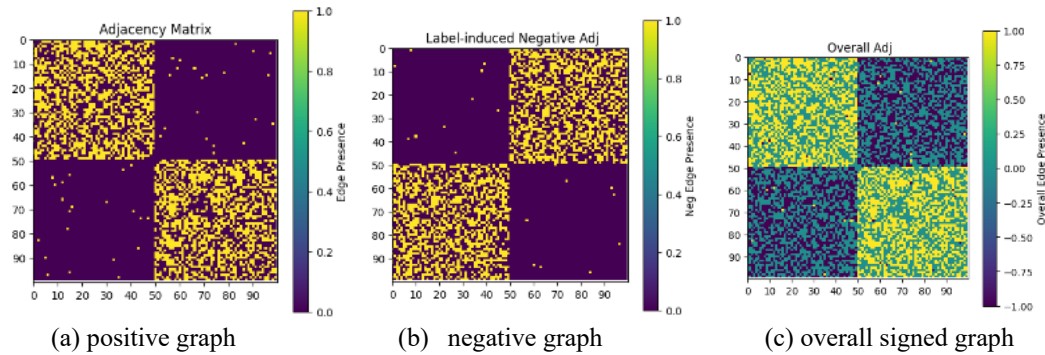

Figure 10: The visualization of the adjacency matrix of Label-SBP. Here left is the positive graph; middle is the negative graph; right is the overall signed graph.

moderately deep layers, as observed in GCN across all datasets. This further confirms the effectiveness of SBP.

### I.4.3   $\beta$ Analysis on Heterophilic Datasets

Our method SBP can outperform other baselines under $\beta = 1$ across different layers, so we do not tune our hyper-parameters carefully. However, since $\beta$ is the weight of the negative adjacency matrix (equation 4.2) representing the repulsion between different nodes, as seen in Figure 4 and 5, the best performance of SBP appears when $\beta$ is larger in the heterophilic graphs, so the result in Figure 3a(a) is not the best performance of our SBP. To further show the effectiveness of our SBP, we conduct experiments on Cornell with different $\beta$ in Table 12, the best $\beta$ is 20 where the performance increases 25 points in deep layer 50.

### I.4.4   SBP on more benchmarks

We further compare our SBP with SGC on six additional datasets [36] in Table 13. Our SBP outperforms SGC on five out of these six datasets. We believe that these six datasets, combined with

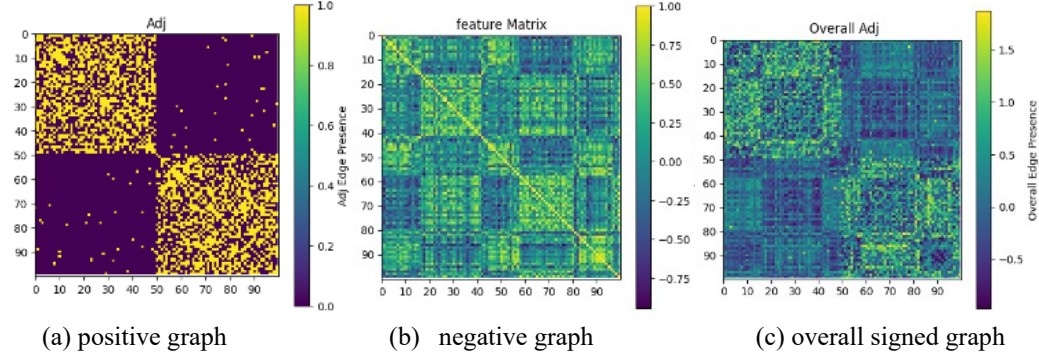

| (a) positive graph | (b)  negative graph | (c) overall signed graph |

Figure 11: The visualization of the adjacency matrix of Feature-SBP. Here left is the positive graph; middle is the negative graph; right is the overall signed graph.

the nine datasets presented in Table 13 of our paper, provide sufficient evidence to demonstrate the effectiveness of our approach.

### I.4.5   Different Backbones

In this paper, we focus on introducing a novel theoretic signed graph perspective for oversmoothing analysis, so we do not take many tricks into account or carefully fine-tune our hyperparameters. Thus, our results in the paper are not as comparable to previous baselines [20, 47, 48]. However, existing oversmoothing researches are indeed hard to compare, because they are often composed of multiple techniques — such as residual, BatchNorm, data augmentation — and the parameters are often heavily (over-)tuned on small-scale datasets. And it becomes clear that to attain SOTA performance, one needs to essentially compose multiple such techniques without fully understanding their individual roles. For example, GCNII uses both initial residual connection and identity map, futher combined with dropout.

Our goal is to provide a new unified understanding of these techniques, so we justified it by showing that SBP as a single simple technique can attain good performance. And we believe that it would work complementarily with other techniques in the field, because oversmoothing is still challenging to solve with a very larger depth.

To further verify the effectiveness, we combine our SBP to one of the SOTA settings GCNII [20] and the results are as seen in Table 14. The results indicate that after combining our method, GCNII demonstrates greater robustness as the layers go deeper, particularly in the middle layers (layer=8), highlighting the efficacy of our signed graph insight.

### I.4.6   SBP **on Large-scale graphs**

We conducted experiments with a larger graph ogbn-products than ogbn-arxiv under 100 epochs and 2 layers in Table 15. The results indicate that our SBP outperforms the initial GCN baselines. Given the results presented for ogbn-arxiv in Table 5 of our paper, we believe these findings adequately demonstrate the performance of our SBP on large-scale graphs.

### I.4.7   Further Optimization

Based on the experiment results, we want to propose 2 strategies for further optimization.

1) hyper-parameter tuning on the negative weight $\beta$. As seen in Figures 4 and 5, we found that $\beta$ influences the performance a lot, our default $\beta = 1$ for Table 2 and 3 is certainly not optimal for the above 4 homophilic datasets. We suggest tuning higher $\beta$ for the heterophilic graphs since they need more repulsion and smaller for the homophilic datasets. As the layer deepens, maybe greater weight should be placed on the negative adjacency graphs to alleviate oversmoothing.

2) adapt our SBP to more effective GNNs. Our method is simple, architecture-free, without additional learnable parameters, and thus can be flexibly applied in various architectures. As seen in

Table 11: GCN test accuracy (%) comparison results. The best results are marked in blue and the second best results are marked in gray on every layer. We run 5 runs for the seed from $0 - 4$ and demonstrate the mean $\pm$ std in the table.

| Model | #L=2 | #L=4 | #L=8 | #L=16 | #L=32 | #L=64 |
|---|---|---|---|---|---|---|
| | | | *Cora* [37] | | | |
| GCN [8] | 80.68 ±0.09 | 79.69 ±0.00 | 74.32 ±0.00 | 30.95 ±0.00 | 30.95 ±0.00 | 24.85 ±7.46 |
| GAT [10] | 81.48 ± 0.48 | 80.69 ± 0.93 | 58.59 ± 1.95 | 25.17 ± 5.67 | 31.93 ± 0.21 | 28.38 ± 0.00 |
| wGCN [46] | 80.97 ± 0.28 | 80.51 ± 0.00 | 80.46 ± 1.77 | 70.53 ± 22.09 | 80.02 ± 0.12 | 27.90 ± 6.09 |
| BatchNorm [41] | 78.09 ±0.00 | 77.87 ±0.02 | 73.62 ±0.57 | 70.79 ±0.00 | 53.90 ±2.19 | 35.32 ±3.41 |
| PairNorm [17] | 79.01 ±0.00 | 78.26 ±0.50 | 73.21 ±0.00 | 62.96 ±0.00 | 48.13 ±0.91 | 44.01 ±3.46 |
| ContraNorm [16] | 81.55 ±0.21 | 79.61 ±0.75 | 77.71 ±0.00 | 63.35 ±0.00 | 44.56 ±4.83 | 38.97 ±0.00 |
| DropEdge [22] | 78.38 ±0.00 | 74.47 ±0.00 | 26.91 ±0.83 | 22.24 ±3.04 | 27.18 ±0.00 | 25.98 ±6.00 |
| Residual | 80.68 ±0.09 | 78.77 ±0.00 | 79.26 ±0.21 | 40.91 ±0.00 | 30.95 ±0.00 | 27.90 ±6.09 |
| Feature-SBP | 80.44 ±0.83 | 79.26 ±1.18 | 78.56 ±0.59 | 77.22 ±0.55 | 73.65 ±0.48 | 61.62 ±5.24 |
| Label-SBP | 80.31 ±0.70 | 79.16 ±1.30 | 79.50 ±0.00 | 77.43 ±1.49 | 74.52 ±0.36 | 65.02 ±2.97 |
| | | | *CiteSeer* [38] | | | |
| GCN [8] | 67.45 ±0.54 | 65.62 ±0.25 | 37.22 ±2.46 | 22.03 ±4.76 | 19.65 ±0.00 | 19.65 ±0.00 |
| GAT [10] | 69.91 ± 0.86 | 67.47 ± 0.22 | 44.71 ± 3.07 | 23.48 ± 1.36 | 24.40 ± 0.40 | 25.95 ± 2.17 |
| wGCN [46] | 66.21 ± 0.63 | 66.49 ± 0.69 | 66.79 ± 0.00 | 57.54 ± 18.94 | 19.65 ± 0.00 | 19.65 ± 0.00 |
| BatchNorm [41] | 63.44 ±0.94 | 62.34 ±0.25 | 61.36 ±0.00 | 50.58 ±1.24 | 41.41 ±0.00 | 35.00 ±1.09 |
| PairNorm [17] | 63.58 ±0.63 | 64.32 ±0.95 | 61.95 ±1.24 | 50.06 ±0.00 | 37.21 ±1.87 | 36.09 ±0.07 |
| ContraNorm [16] | 66.83 ±0.49 | 64.78 ±0.92 | 60.70 ±0.60 | 44.79 ±1.65 | 37.36 ±0.25 | 30.85 ±0.81 |
| DropEdge [22] | 63.86 ±0.03 | 62.24 ±0.90 | 24.73 ±5.72 | 20.65 ±0.00 | 20.04 ±0.19 | 19.95 ±0.09 |
| Residual | 67.45 ±0.54 | 66.21 ±0.16 | 67.34 ±0.00 | 33.21 ±0.00 | 19.65 ±0.00 | 19.65 ±0.00 |
| Feature-SBP | 67.38 ±0.66 | 66.94 ±0.00 | 66.29 ±0.02 | 65.35 ±1.99 | 61.43 ±0.00 | 42.09 ±1.65 |
| Label-SBP | 67.23 ±0.64 | 66.72 ±0.00 | 66.29 ±0.89 | 65.50 ±2.13 | 59.93 ±0.85 | 44.41 ±1.57 |
| | | | *PubMed* [39] | | | |
| GCN [8] | 76.44 ±0.34 | 76.52 ±0.32 | 69.58 ±5.89 | 39.92 ±0.00 | 39.92 ±0.00 | 39.92 ±0.00 |
| +BatchNorm [41] | 75.52 ±0.12 | 77.15 ±0.00 | 77.10 ±0.00 | 76.92 ±0.00 | 75.43 ±0.00 | 69.33 ±1.01 |
| +PairNorm [17] | 75.66 ±0.11 | 76.71 ±0.00 | 77.99 ±0.00 | 77.22 ±0.39 | 75.52 ±2.02 | 71.22 ±3.68 |
| +ContraNorm [16] | 76.05 ±0.33 | 78.42 ±0.00 | OOM | OOM | OOM | OOM |
| +DropEdge [22] | 73.41 ±0.03 | 73.96 ±0.79 | 52.51 ±10.91 | 40.27 ±0.00 | 39.90 ±0.59 | 40.08 ±0.39 |
| +Residual | 76.44 ±0.34 | 77.28 ±0.00 | 77.38 ±0.00 | 63.14 ±3.05 | 39.92 ±0.00 | 39.92 ±0.00 |
| Feature-SBP | 75.72 ±0.06 | 76.84 ±0.00 | 78.39 ±0.00 | 79.71 ±0.00 | 77.59 ±0.23 | 78.06 ±0.13 |
| Label-SBP | 76.33 ±0.25 | 76.91 ±0.00 | 77.60 ±0.49 | 76.31 ±0.00 | 77.17 ±0.67 | 78.01 ±0.16 |

Table 12: Ablation study of negative weight $\beta$ on Cornell dataset.

| Layer | 2 | 5 | 10 | 20 | 50 |
|---|---|---|---|---|---|
| $\beta = 0.1$ | 72.97 ± 0.00 | 67.57 ± 0.00 | 51.53 ± 0.00 | 35.14 ± 0.00 | 29.73 ± 0.00 |
| $\beta = 1$ (default) | 72.97 ± 0.00 | 67.57 ± 0.00 | 51.53 ± 0.00 | 45.95 ± 0.00 | 35.14 ± 0.00 |
| $\beta = 10$ | 70.27 ± 0.00 | 67.57 ± 0.00 | 58.11 ± 1.35 | 51.53 ± 0.00 | 51.53 ± 0.00 |
| $\beta = 20$ (best) | 70.27 ± 0.00 | 70.27 ± 0.00 | 67.57 ± 0.00 | 59.46 ± 0.00 | 59.46 ± 0.00 |
| $\beta = 50$ | 64.60 ± 0.00 | 40.54 ± 0.00 | 40.54 ± 0.00 | 40.54 ± 0.00 | 40.54 ± 0.00 |

Appendix I.4.5, we adapt our SBP to the GCNII models, and the results increase more than adaptation in vanilla GNN as shown in Table 2 and 3. Besides, compared to the GCNII, our SBP is more robust and stable to the layers as seen in Table 14.

Table 13: Performance Comparison on more datasets

| | actor | penny94 | roman-empire | Tolokers | Questions | Minesweeper |
|---|---|---|---|---|---|---|
| SGC | 29.18 ± 0.10 | 72.56 ± 0.05 | 40.83 ± 0.03 | 78.18 ± 0.02 | 97.09 ± 0.00 | 80.43 ± 0.00 |
| Feature-SBP | 34.93 ± 0.02 | 75.68 ± 0.01 | 66.48 ± 0.02 | 78.24 ± 0.04 | 97.14 ± 0.02 | 80.00 ± 0.00 |
| Label-SBP | 34.94 ± 0.00 | 75.74 ± 0.01 | 66.32 ± 0.01 | 78.46 ± 0.08 | 97.15 ± 0.02 | 80.00 ± 0.00 |

Table 14: Performance Comparison between SBP and GCNII under the GCNII settings on Cora and Citesser datasets

| | | 2 | 4 | 8 | 16 | 32 | 64 |
|---|---|---|---|---|---|---|---|
| | GCNII | $78.58 \pm 0.00$ | $77.76 \pm 0.24$ | $73.47 \pm 3.82$ | $78.12 \pm 1.32$ | $82.54 \pm 0.00$ | $81.34 \pm 0.53$ |
| Cora | Label-SBP | $78.74 \pm 1.54$ | $78.87 \pm 0.00$ | $79.14 \pm 0.35$ | $79.17 \pm 0.41$ | $80.86 \pm 0.32$ | $81.38 \pm 0.30$ |
| | Feature-SBP | $77.95 \pm 0.91$ | $78.82 \pm 0.00$ | $78.11 \pm 1.62$ | $78.82 \pm 0.29$ | $81.82 \pm 0.47$ | $81.65 \pm 0.40$ |
| | GCNII | $61.66 \pm 0.67$ | $63.23 \pm 2.31$ | $64.58 \pm 2.66$ | $66.21 \pm 0.64$ | $69.38 \pm 0.83$ | $69.73 \pm 0.26$ |
| Citesser | Label-SBP | $65.31 \pm 0.63$ | $63.93 \pm 3.66$ | $68.33 \pm 0.99$ | $66.46 \pm 0.00$ | $70.00 \pm 0.81$ | $69.47 \pm 0.25$ |
| | Feature-SBP | $65.63 \pm 0.87$ | $64.43 \pm 3.55$ | $68.44 \pm 1.19$ | $66.94 \pm 0.00$ | $69.98 \pm 0.93$ | $69.66 \pm 0.28$ |

Table 15: Performance of different models on ogbn-products dataset. Time means the runtime, the format is (hour: minutes: seconds).

| Method | Accuracy | Time |
|---|---|---|
| GCN | 73.96 | 00:06:33 |
| BatchNorm | 74.93 | 00:06:18 |
| Feature-SBP | 74.90 | 00:06:43 |
| Label-SBP | 76.62 | 00:06:39 |

