# OpenReview forum: "A Signed Graph Approach to Understanding and Mitigating Oversmoothing"
_NeurIPS.cc/2025/Conference — NeurIPS 2025 poster_

### Official Review · Reviewer_WoFN · 2025-06-24

**Clarity:** 3
**Significance:** 3
**Originality:** 4
**Rating:** 5
**Confidence:** 4

**Summary:**

Oversmoothing is a well-known problem that plagues GNNs as increasing the number of layers leads to the indistinguishability of features indicating collapse (or consensus) to a single feature (or label) across nodes. This inhibits the use of deep GNNs in practice and a number of strategies have been proposed to ameliorate this problem.

This article provides a different and novel strategy: it uses a unified analytical framework based on the theory of signed graphs. Oversmoothing is defined as a consensus where all node features collapse to the average in the infinite depth limit. This is dynamically characterized by a structured balance (between positive (attractive) and negative (repulsive)) edges and crisp boundary in terms of a parameter $\beta$ provided in Th. 3.2. This notion of structured balance arises in social network analysis and its use in this context is novel. Many existing techniques (dropout, normalization, residuals) are shown to implicitly introduce negative edges.

The authors go beyond just analysis of existing techniques and propose SBP, a plug- and play module to construct signed graphs promoting structural balance, which can be used with any GNN backbone, with positive edges for connecting similar nodes (intra-class) and negative edges for dissimilar nodes (inter-class). 2 variants (Label-SBP and Feature-SBP) are proposed.

The authors test their method on 9 datasets, both homophilic and heterophilic, and show that their method does better than traditional ways to ameliorate oversmoothing. In particular, the use of very deep graphs (300 layers) and the resulting lack of deterioration of performance is noteworthy.

**Questions:**

Please address the questions mentioned in the Weaknesses section. In particular, the questions on continuous metrics for over smoothing, comparison with SOTA, task diversity and scaling.

**Ethical Concerns:**

["NO or VERY MINOR ethics concerns only"]

**Final Justification:**

As my detailed review and the authors' rebuttal and discussions show, this article is a significant contribution to the vexing problem for over smoothing for GNNs. It certainly does not resolve all the issues and its assumptions are strong as far as the theory is concerned. Moreover, the experiments for very large graphs are limited. However, the contribution is solid and there is much to build on.

**Limitations:**

Please see the weaknesses section. I am happy to raise the score if some of the weaknesses are addressed.

**Quality:**

3

**Strengths And Weaknesses:**

Strengths:

-- Addressing an important problem in GNNs.

--- Introduces a novel framework for over smoothing, based on signed graphs and structured balance. While signed graphs have been used in the context of heterophilic datasets, their explicit use in this setting appears to be totally novel.

--- Crisp criterion for over smoothing in terms of consensus.  Thrm 3.2 provides an exact phase transition on when over smoothing occurs and when it does not.

---  SBP is a new plug-and-play module to overcome over smoothing.

--- Provides decent empirical benchmarking and showcases the use of very deep GNN layers.

--- Very interesting addition to the GNN literature by introducing a different framework for understanding over smoothing.

Weaknesses:

--- Theory assumes exact structural balance in a signed graph. Is this even possible for real-world data ? Can the authors at least comment on whether this criterion is satisfied for the datasets that they consider. They mention weak-balance but do not analyze it here.

--- This reviewer's main theoretical concern is with the binary over smoothing/non-oversmoothing nature of the consensus criterion. If you compare with other criteria such as the decay in Dirichlet energy or Cosine similarity (see Ref.[49] for a detailed survey), they clearly provide a layer-dependent rate and an easily computable criterion for measuring over-smoothing. The authors' metric is qualitative and they only use accuracy as a proxy for the extent of over smoothing. This lack of a quantification for over-smoothing is the key limiting factor in practical applications.

--- SBP is a fixed signed edge construction. The signs are assigned and cannot be altered, learned or adapted. This might be a straitjacket for the method. In particular, the graph is modified prior to the training and there is no mechanism to alter it during training.

-- The paper does not measure over smoothing through numerous criteria that have been proposed for it -- such as Dirichlet energy ? Cosine Similarity ? Gradient norms ? Just using accuracy as a proxy for oversmoothing is not entirely convincing. The authors should at least Dirichlet energy as a function of width to see if the method is able to actually overcome over smoothing or not.

-- The experimental evaluation is limited. In particular, only traditional methods such as edge-dropout, normalizations, residual connections are considered. What is missing is comparison with state of the art methods to overcome over smoothing. For instance, G2 (arXiv:2210.00513) is very strong baseline as it has been rigorously shown to prevent collapse of Dirichlet energy and hence, over smoothing and it can scale to very large graphs (Snap-patents, 3M nodes, 13M edges). A comparison with the state of the art is necessary. Moreover, SIGN, GPR-GNN, and other adaptive propagation models can be compared with to justify empirical performance.

--- The backbones that the paper proposes are limited to GCN variants and other classes of more frequently used backbones such as (variants of) GAT and GraphSAGE are missing. Perhaps, it is essential to use atleast GAT as a backbone in one of the datasets.

--- The task diversity is very limited, only to transductive node classification. This is too specific -- the authors could other tasks, link prediction, molecular property prediction or PPI , inductive node classification etc.

--- Scaling: the authors only showcase performance on limited size datasets with Obgn-Arxiv being the largest. Scaling to something like snap-patents, a very heterophilic graph  with 3M nodes and 13M edges would be a very interesting challenge for the method. There is a fundamental issue in terms of scaling as the Feature-SBP has a dense similarity matrix (quadratic scaling in nodes) and although the authors mention low-rank approaches, they do not consider it in any detail. The reviewer's concern is that SBP will not scale to large scale graphs.

---

> ### Author Rebuttal · Authors · 2025-07-30
>
> We thank the reviewer for taking the time to review our paper and providing constructive feedback. In what follows, we provide detailed responses to the comments raised by the reviewer. For clarity, we use the following notations:
>
> - W – Weakness
> - Q - Question
>
> **W1 How to apply structural balance to real-world data.**
>
> ---
>
> Thank you for the question. We address this in Appendix H, where we introduce the Structural Imbalance Degree (SID), as defined in Definition H.1, to quantify structural balance in real-world data. As shown in Table 6, our SBP method consistently achieves lower SID scores compared to other anti-oversmoothing baselines, indicating its effectiveness in enhancing structural balance.
>
> **W2 & W4 Lack of metrics for measuring oversmoothing.**
>
> ---
>
> Thank you for your comment. We also used the Dirichlet energy to measure oversmoothing. Furthermore, in Appendix F, we theoretically demonstrate that structural balance mitigates oversmoothing from the perspective of Dirichlet energy even in the infinite-depth regime, highlighting the robustness of our approach.
>
> **W3 SBP is a fixed signed edge construction.**
>
> ---
>
> Thank you for your insightful comments. We appreciate the suggestion to extend the positive $\alpha$ and negative weights $\beta$ as learnable parameters, and we will consider this for future work. In the current study, we modify the graphs based on prior information to create more structurally balanced graphs, which helps alleviate oversmoothing. Our goal is to demonstrate the effectiveness of our unified signed graph framework. We look forward to exploring learnable parameters in subsequent research.
>
>
>
> **W5 & W6 Comparison with SOTA/different backbones**
>
> ---
>
> Thank you for the suggestions! We compare our SBP with the aforementioned methods on the Cora dataset. As shown better, SBP achieves better performance.
>
> | Methods | acc |
> | --- | --- |
> | g2 | 78.11  |
> | sign | 69.3 |
> | gpr-gnn | 79.51 |
> | gat | 81.48 |
> | GraphSAGE | 70.89 |
> | feature-sbp | 82.46 |
> | label-sbp | **82.90** |
>
> We also adapt our SBP method to GAT backbones on the Cora dataset as suggested. As shown in the results below, SBP remains effective with the GAT backbone.
>
> | Methods | acc |
> | --- | --- |
> | gat | 81.48 |
> | label-sbp(gat) | **83.31** |
> | feature-sbp(gat) | 82.59 |
>
> We will add these experiments in the updated manuscript. Thank you for the suggestions!
>
> [1] Gradient Gating for Deep Multi-Rate Learning on Graphs
>
> [2] SIGN: Scalable Inception Graph Neural Networks
>
> [3] Adaptive Universal Generalized PageRank Graph Neural Network
>
> **W7 Task diversity**
>
> ---
>
> Thank you for the comment. We have evaluated our SBP method on the link prediction task, and the results on Cora are shown below. SBP achieves better  accuracy, demonstrating its effectiveness beyond node classification.
> | Methods | acc |
> | --- | --- |
> | gcn | 0.8389 |
> | feature-sbp | 0.8430 |
> | lable-sbp | **0.8617** |
>
> **W8 Scaling to large-sized graphs.**
>
> ---
>
> Thank you for the comment. We conduct experiments on the **ogbn-products** dataset, which is comparable in scale to **snap-patents** and contains even more edges.
>
> |  | nodes | edges |
> | --- | --- | --- |
> | snap-patents | 3,774,768 | 16,518,948 |
> | ogbn-products | 2,449,029 | 61,859,140 |
>
> The results, presented in Appendix I.4.6 (Table 15), show that our SBP method outperforms the baselines, highlighting its scalability.
>
> | Methods | acc | time |
> | --- | --- | --- |
> | gcn | 73.96 | 00:06:33 |
> | batchnorm | 74.93 | 00:06:18 |
> | feature-sbp | 74.90 | 00:06:43 |
> | label-sbp | **76.62** | 00:06:39 |
> --------------------
> We appreciate your questions and comments very much. Please let us know for any further questions.

---

> > ### Comment · Reviewer_WoFN · 2025-08-01
> > **Reply to Authors**
> >
> > I thank the authors for addressing several of my questions satisfactorily. In particular, Appendix F was very useful. I am still to be convinced about 1 point though: given the binary nature of your over smoothing criteria (modulo the link with Dirichlet energy in the asymptotic regime), I would like you to empirically explore an example where the Dirichlet energy does not decay when your SBP approach is added whereas the vanilla GNN shows a decay of Dirichlet energy as the number of layers is increased ? Can you perform such an experiment if possible ? Without that, you are still using accuracy as the proxy (albeit a good one) for over smoothing rather than a direct over smoothing metric ?

---

> > > ### Author Response · Authors · 2025-08-01
> > > **Thanks!**
> > >
> > > Thank you for your quick response. We are very glad to hear that you find our rebuttal satisfactory.
> > >
> > > In line with your new suggestion, we measured the Dirichlet energy on the Cora dataset across various numbers of layers (2-100) using both the vanilla SGC and our methods, Feature/Label-SBP. The results show that the Dirichlet energy of the vanilla GNN decreases from 0.1362 at layer 2 to 0.0109 at layer 100 as the number of layers increases. After incorporating our SBP approach, the Dirichlet energy remains stable at 0.1452 even at the deep layer 100, aligning with our analysis of the oversmoothing theorem.
> > >
> > > | methods | L=2 | L=10 | L=50 | L=100 |
> > > | --- | --- | --- | --- | --- |
> > > | sgc | 0.1362 | 0.0164 | 0.0114 | 0.0109 |
> > > | label-sbp | 0.1362 | 0.1891 | 0.1452 | 0.1452 |
> > > | feature-sbp | 0.1362 | 0.1891 | 0.1452 | 0.1452 |
> > >
> > > The Dirichlet energy metrics align well with the node classification accuracy of these methods, particularly in the deeper layers (100). In that case, the vanilla SGC experiences oversmoothing, which is reflected in its low Dirichlet energy and leads to a significant drop in accuracy from 80.32 to 16.80. In contrast, our SBP approach maintains a high accuracy of 80.17.
> > >
> > > Hope you find this new experiment satisfactory! Thank you for the insightful comment, which helps strengthen our work. We will add these new results to the revision. Please let us know if there is more to clarify and discuss.

---

> > > > ### Comment · Reviewer_WoFN · 2025-08-01
> > > > **Response to Authors.**
> > > >
> > > > I thank the authors for their prompt reply. This experiment provided a clear answer to my concern. The Dirichlet energy and accuracy were highly correlated. This further reinforces my point that this paper contributes significantly to the issue to over smoothing for GNNs. I recommend acceptance.

---

### Official Review · Reviewer_xGFU · 2025-07-02

**Clarity:** 2
**Significance:** 2
**Originality:** 2
**Rating:** 5
**Confidence:** 5

**Summary:**

The author addresses the common oversmoothing problem in deep graph neural networks (GNNs) by proposing a unified perspective of "signed graphs" to reinterpret existing alleviation strategies such as normalization, residual connections, and random edge deletion. The paper first proves that these methods are essentially equivalent to implicitly adding negative edges in the graph, thus introducing an "attraction - repulsion" mechanism during information propagation. Subsequently, the author points out that simply adding negative edges is still insufficient to completely solve oversmoothing; the truly crucial factor is the "structured organization" of positive and negative edges. Based on the structural balance theory, the author proposes a Structural Balanced Propagation (SBP) that requires no additional learnable parameters and can be plugged in directly. SBP has two implementations:

1. Label - SBP — Using known labels, positive edges are set between nodes of the same class, and negative edges are set between nodes of different classes.
2. Feature - SBP — In scenarios with scarce labels, the signs of edges are automatically inferred based on feature similarity.

Theoretically, the author provides sufficient conditions for SBP to suppress oversmoothing and proves that node representations on balanced signed graphs will converge to class - specific steady - state values. Experiments demonstrate the superiority of SBP on 9 homogeneous/heterogeneous datasets with a depth of 300 layers, and a sparsified version is provided for large - scale graphs.

**Questions:**

View the Weaknesses in Strengths And Weaknesses.

**Ethical Concerns:**

["NO or VERY MINOR ethics concerns only"]

**Final Justification:**

5

**Limitations:**

yes

**Paper Formatting Concerns:**

No paper formatting concerns.

**Quality:**

3

**Strengths And Weaknesses:**

Strengths
- Unified theoretical perspective: It subsumes various known techniques as "implicit negative edges", provides clear mathematical derivations, and offers novel theoretical insights.
- Comprehensive experiments: Covering 9 benchmarks, homogeneous and heterogeneous graphs, ultra-deep layers (300 layers), and the large-scale OGBN-ArXiv dataset, with consistently leading results.

Weaknesses
- Idealized theoretical assumptions: The key convergence theorem relies on assumptions such as linear propagation, complete graphs, or bipartite clustering, which still have a gap from real sparse graphs.
- Parameter - sensitivity: Performance is highly sensitive to the negative edge weight β. The paper only uses grid search, lacking an automatic or adaptive adjustment mechanism.
- Risk of graph densification: By default, SBP requires constructing n² - level negative edges, resulting in significant memory consumption for large graphs. Although the v2 scheme improves this, the accuracy slightly decreases.
- Insufficient comparison with the latest methods: There is no systematic comparison with recent anti-oversmoothing or heterogeneity methods.

---

> ### Author Rebuttal · Authors · 2025-07-30
>
> We appreciate that you found our theory clear and our experiments comprehensive. We value the opportunity to address your concerns and provide clarification in this rebuttal. Please find our detailed responses to your specific questions below. For clarity, we use the following notations:
>
> - W – Weakness
> - Q - Question
>
>
> **W1 The theorem relies on assumptions such as linear propagation, complete graphs, or bipartite clustering, which still have a gap from real sparse graphs.**
>
> ---
>
> We understand that you have concerns about the generality of the analysis, but in fact, our analysis can be easily extended to broader scenarios that align with practice:
>
> - Linear propagation: Our current linear activation ($\sigma(x) = x$) is generalized to a bounded activation $f(Z)_c$ as described in Theorem 4.2. It could be further extended to the monotonically increasing activation function.
>
> - Complete graphs: The purpose of having $\mathcal{G}$ as a complete graph is to ensure connectivity. This requirement can be relaxed to the number of connected components $k(\mathcal{G}^+) > 2$, with $\mathcal{G}^-$ containing at least one negative edge, generalized to the graph consisting of multiple disjoint subsets, thus capturing a wider variety of scenarios. This modification satisfies the constraints of Lemma D.3 and supports the proof of Theorem 4.2 in Appendix D.
>
> - Bipartite clustering: We have expanded our approach to multiple clusters, namely weakly structual balance, in Remark 4.3, see detailed theorems and proofs in Appendix G.
>
> We will add the discussion about the assumptions in the revision.
>
> **W2 Performance is sensitive to negative weight $\beta$, lacking an automatic or adaptive adjustment mechanism.**
>
> ---
>
>
> In our experiments, we empirically observe that setting $\beta = 0.9$ consistently yields strong performance across both homogeneous and heterogeneous graphs, making it a reasonable default choice.
>
> Moreover, we provide a detailed sensitivity analysis of $\beta$ in Section 5.3. Specifically, we vary $\beta$ from 0.1 to 100 and observe consistent trends: for heterogeneous graphs, a larger $\beta$ (e.g., 10) improves performance, whereas for homogeneous graphs, a smaller $\beta$ (e.g., 0.1) is more effective (see lines 303–313). This analysis highlights the impact of $\beta$ and can guide practitioners in selecting appropriate values depending on the graph type.
>
> **W3 SBP results in significant memory consumption for large graphs.**
>
> ---
> We understand that you are worried about the memory consumption of our SBP. To make it scalable, we have improved the next version (v2) of our Label/Feature-SBP. Among them, Label-SBP-v2 is very scalable (even enhances the sparsity of the natural graph structure) and performs well, as shown in Table 3, particularly at deeper layers (16), addressing the memory issues.
>
> **W4 Insufficient comparison with the latest methods.**
>
> ---
>
> Thank you for your comment. In response to your suggestion, we have compared our method to the latest baselines graff[1] and g2[2] recommended by Reviewer WoFN and mzv3 on Cora. Our SBP demonstrates higher accuracy than these baselines.
>
> | methods | acc |
> | --- | --- |
> | graff[1] | 80.91 |
> | g2[2]  | 78.11  |
> | feature-sbp | 82.46 |
> | label-sbp | **82.90** |
>
>
> Nevertheless, we would like to emphasize that our work is not solely focused on achieving SOTA performance; its primary contribution lies in the unified theoretical framework we introduce. This framework offers a novel signed perspective that connects and generalizes existing over-smoothing mitigation methods, providing deeper insights into their underlying mechanisms.
>
> [1] Understanding Convolution on Graphs via Energies (TMLR 2023)
>
> [2] Gradient Gating for Deep Multi-Rate Learning on Graphs
>
> ---------------
> We appreciate your questions and comments very much. Please let us know for any further questions.

---

> > ### Comment · Reviewer_xGFU · 2025-08-04
> >
> > Thank you for the reviewer’s response. I have raised my score to 5. Good luck!

---

### Official Review · Reviewer_3cVw · 2025-07-02

**Clarity:** 3
**Significance:** 3
**Originality:** 3
**Rating:** 4
**Confidence:** 3

**Summary:**

This paper proposes a method based on signed graphs to understand and mitigate the over-smoothing problem in Graph Neural Networks (GNNs). Through the framework of signed graphs, the authors demonstrate that various existing techniques for mitigating over-smoothing can be interpreted as implicitly introducing negative edges. Furthermore, the authors propose the Structural Balance Propagation (SBP) method, which enhances structural balance in the constructed signed graph by assigning signed edges based on either labels (label-SBP) or feature similarity (feature-SBP). For large-scale graphs, they also introduce Label-SBP-v2. Experiments show that SBP effectively improves classification accuracy and alleviates over-smoothing in deep GNNs with up to 300 layers. This method provides a theoretical explanation for existing over-smoothing mitigation methods and offers a new direction for the design of signed message-passing in deep GNNs.

**Questions:**

1. Despite being supported by structural balance theory, can SBP be further optimized to adapt to more complex graph structures?

2. The experimental settings for large-scale graphs are not sufficiently detailed. Could you provide more details to support the applicability of SBP on large-scale graphs? Additionally, could you provide experimental data on larger-scale graphs and differentiate the experiments for settings such as large-scale vs. small-scale graphs and homogeneous vs. heterogeneous graphs?

3. The paper lacks experiments on GCNs with deep layers for heterogeneous graphs. The number of experiments on heterogeneous graphs is notably less than on homogeneous graphs.

4. The experiments lack a detailed description of F(z)c​ (e.g., the range of values for c).

**Ethical Concerns:**

["NO or VERY MINOR ethics concerns only"]

**Final Justification:**

The authors have addressed my questions, so I will maintain my positive score.

**Limitations:**

The authors have adequately discussed the limitations of their work, particularly regarding the scope of the method's evaluation and its applicability to large-scale graphs.

**Paper Formatting Concerns:**

The phrase "Table ??" appears multiple times in the paper, indicating missing references to table numbers.

The names of the test datasets should be included either in the text or in the figures/tables. This issue is present in numerous figures and tables.

**Quality:**

3

**Strengths And Weaknesses:**

1. The proposed SBP method is introduced clearly, is easy to understand, and demonstrates good scalability and adaptability.

2. The theoretical analysis is solid and strongly supports the proposed method, offering a new perspective for understanding the over-smoothing problem in GNNs.

3. Empirical results on various datasets show that the proposed method effectively improves performance and mitigates over-smoothing under multiple settings.

4. Several details about the proposed method are not sufficiently clear. For instance, the analysis of its adaptability to large-scale graphs could be more detailed.

5. There is a lack of illustrative diagrams to show the propagation process of SBP.

6. The discussion on the applicability of the method to different types of graphs (homogeneous and heterogeneous graphs) could be more in-depth.

---

> ### Author Rebuttal · Authors · 2025-07-30
>
> Thank you for taking the time to review our paper and providing constructive feedback. Below, we provide individual responses to your comments and questions. For clarity, we use the following notations:
>
> - W – Weakness
> - Q - Question
>
> **W4 The analysis of its adaptability to large-scale graphs could be more detailed.**
>
> ---
>
>
> We present the detailed formulation and implementation of SBP-v2, which is specifically designed to scale effectively with large graphs. In Appendix I.3 of ogbn-arxiv, SBP achieves a performance score of 69.53 at layer 16, surpassing GCN's score of 59.13. Additionally, in Table 15 of ogbn-products, SBP attains a score of 76.62, outpacing the Baseline BatchNorm's score of 74.93. Appendix I.3 includes key equations and practical considerations for applying SBP-v2 in large-scale settings.
>
>
> **W5 Lack of illustrative diagrams to show the propagation process of SBP.**
>
> ---
>
> Thank you for your comment. We have included visualizations of node features from layer 0 to 200 in Figures 8 and 9 of Appendix I. These figures demonstrate that the node features induced by baseline SGC converge to the same clusters as the layer increases, while our SBP's node features converge to distinct clusters even under layer 200. We will add more illustrations in the revision to enhance the explanation of our SBP evolution.
>
> **W6 The discussion on the applicability of the method to different types of graphs (homogeneous and heterogeneous graphs) could be more in-depth.**
>
> ---
>
> That is a good question. Our SBP can be tailored to the hyperparameters based on the homophily level. In Table 12 on the heterogeneous graph Cornell, we enlarge the negative weight $\beta$ of SBP, and it performs even better than the small ones. Follow your suggestions, we in-depth scale from 0.1 to 100 the negative weight with the controllable homophily level in Figure 4, and we find that the accuracy increases with beta in heterogeneous graphs and decreases in homogeneous graphs. These results further highlight the role of $\beta$ as a repulsive force in the message-passing process, supporting our signed graph perspective for understanding and mitigating oversmoothing. We have included the detailed discussion in the revision.
>
> **Q1 Can SBP be further optimized to adapt to more complex graph structures?**
>
> ---
>
> Our SBP can be adapted to arbitrary graph structures, provided that informative node features or node labels are available to construct a structurally balanced framework during propagation. However, if the node labels contain significant noise or if the node features are unhelpful or harmful, our method may become ineffective. We will add more discussion in the next version.
>
> **Q2 Could you provide more details to support the applicability of SBP on large-scale graphs? Distinguish different graph settings.**
>
> ---
>
>  We conducted large-scale experiments on the ogbn-arxiv and ogbn-products datasets, which contain millions of nodes and edges, as summarized as follows.
>
>    | Datasets       | Nodes      | Edges        | Type  |
> |----------------|------------|--------------|-------|
> | Cora           | 2,708      | 5,429        | Small |
> | OGBN-Arxiv     | 3,774,768  | 16,518,948   | Large |
> | OGBN-Products   | 2,449,029  | 61,859,140   | Large |
>
> The results for the ogbn-product dataset are detailed in Appendix I.4.6 (Table 15), while the results for ogbn-arxiv are presented in Table 3. Our SBP method consistently outperforms the baselines, demonstrating its scalability. For clarity, we include Table 15 below:
>
>   | Method        | Accuracy | Time      |
> |---------------|----------|-----------|
> | GCN           | 73.96    | 00:06:33  |
> | BatchNorm     | 74.93    | 00:06:18  |
> | Feature-SBP   | 74.90    | 00:06:43  |
> | Label-SBP     | **76.62**| 00:06:39  |
>
>
> Homophilic and heterophilic data are classified based on their homophily levels (as defined in Equation H I.1). A high homophily level (>0.7) indicates homophilic data, while a low level (<0.4) indicates heterophilic data. The homophily levels for each dataset are summarized in Table 1. Our findings indicate that SBP performs well across both homophilic and heterophilic datasets.
>
>
>  | Dataset   | Nodes  | Edges  | Homophily Level |
> |-----------|--------|--------|------------------|
> | Cora      | 2,708  | 5,429  | Homo             |
> | Squirrel  | 198,493| 2,089  | Heter            |
>
>
> **Q3 Lack of heterogeneous graphs results.**
>
> ---
>
> Thank you for the comment.
>
> Due to space constraints, we have included the results on heterogeneous graphs in the appendix (Tables 12 and 13), where we evaluate SBP on six additional heterogeneous datasets (Actor, Penny94, Roman-Empire, Tolokers, Questions, and Minesweeper) across 2 to 50 layers. Our SBP achieves results that are on par with, or better than, the baselines.
>
> We will clarify this placement in the revision.
>
> **Q4 The experiments lack a detailed description of $F_C(Z)$**
>
> ---
>
> Thank you for the comment.
>
> - **Theoretical:** As discussed in lines 178–180, $F_C(Z)$ refers to any function that ensures boundedness and prevents divergence in node features.
> - **Experimental:** As noted in line 224, we implement $F_C(Z)$ using layer normalization.
>
> We will clarify this more explicitly in the revision.
>
> -------
> We appreciate your questions and comments very much. Please let us know for any further questions.

---

### Official Review · Reviewer_mzv3 · 2025-07-02

**Clarity:** 3
**Significance:** 3
**Originality:** 3
**Rating:** 5
**Confidence:** 4

**Summary:**

This paper studies oversmoothing from the perspective of structural balance and proposes constructing signed graphs based on node labels or feature similarity to introduce  positive and negative edges that connect similar and dissimilar nodes, respectively, to enable better clustering/classification of nodes from which both homophilic and heterophilic inputs can benefit with the apprpriate setting of hyperparameters.

**Questions:**

Questions:
1. Line 122: What is meant by 'non-trivial negative edges'?
2. Is $A^+=A$ necessary? The notation is a little confusing as the statement on line 135 about beta=0 resulting in the model degenrating into standard unsigned propagation is only true in this case. If inter-class egdes are removed in Label-SBP-v2, is $A^+ \neq A$? Along similar lines, why are the alpha values fixed to 1 and not varied? Even in the case $A^+=A$, could it not be possible that the existing graph is not always fully useful [4,5,6] and thus its contribution can be lowered?
3. In Figure 3(b), how do the baselines perform at these lower label rates? This would highlight the gain of Label-SBP over them in high label rates particularly, and of Feature-SBP in low label rates.
4. Was Feat-SBP-v2 also evaluated on OGB-Arxiv dataset? The results are missing in Table 5.
5. Has the effect of ablating the residual connection the strength of which is controlled by $\gamma$ been studied? How much of SBP's success in alleviating over-smoothing can be attributed to the signed graph component (particularly in deeper layers) than the residual component?
6. Is the result of Feature-SBP for #$L=2$ a typo in Table 10?
7. Is SBP  applicable to only node classification or could it also cater to other graph tasks such as link prediction?

[4] GNNs Use Graphs When They Shouldn't (ICML 2024)

[5] GATE: How to Keep Out Intrusive Neighbors (ICML 2024)

[6] When Are Graph Neural Networks Better Than Structure-Agnostic Methods? ICBINB (NeurIPS 2022)

**Ethical Concerns:**

["NO or VERY MINOR ethics concerns only"]

**Final Justification:**

Given the strengths of the paper listed in the original review, and adequate answers provided by the authors to my questions during the rebuttal, I recommend acceptance of this paper.

**Limitations:**

yes

**Quality:**

3

**Strengths And Weaknesses:**

Strengths:

1. Several existing strategies to mitigate oversmoothing that were developed individually and intuitively are unified under the framework of signed graphs which further elucidates their underlying principles.
2. The study is quite comprehensive - identifies a unifying perspective on existing techniques, analyzes the conditions under which signed graphs optimally benefit node classification, addresses scalability problems, considers both homophilic and heterophilic graphs, extensive experiments across varying datasets, baselines and GNN backbones for prposed SBP method.
3. I particularly appreciate the position of the paper as one that aims to provide more understanding of the matter through theoretical and empirical analysis rather than chasing a SOTA performance using various 'tricks'. This could potentially provide the groundwork for further developing SBP-based techniques.

Weaknesses:
1. The hyperparameter beta as an essential component is most likely difficult to tune, specially since it is unbounded.
2. Albiet not rooted in structural balance theory, other approaches such as GRAFF[1], $\omega$GAT[2], FAGCN[3] that also use signed representations on the original graph structure itself to improve attraction and repulsion between similar and dissimilar nodes, respectively, are not discussed. Could the authors comment on how they are related to SBP and perhaps provide a comparison to see what the more effective approach to using signs to model attraction and repulsion is?
3. Minor: readability is compromised by several missing references in the appendix. (e.g. lines 551, 846, 898, 926, 933).


[1] Understanding Convolution on Graphs via Energies (TMLR 2023)

[2] Improving Graph Neural Networks with Learnable Propagation Operators (ICML 2023)

[3] Beyond Low-frequency Information in Graph Convolutional Networks (AAAI 2021)

---

> ### Author Rebuttal · Authors · 2025-07-30
>
> We are very encouraged by your positive assessment of our work. Below we provide detailed answers to your comments and questions. For clarity, we use the following notations:
>
> - W – Weakness
> - Q - Question
>
> **W1 The hyperparameter $\beta$ is difficult to tune.**
>
> ---
>
> Thank you for your comment.
>
> In Table 1 settings, $\beta$ was chosen as the best from {$0.1, 0.5, 0.9$}. Our SBP performs best on 7 out of 8 graphs across both homogeneous and heterogeneous graphs. We suggest using 0.9 as the default and trying 0.1 for homogeneous graphs, while a larger value, such as 10, may be more effective for heterogeneous graphs, following our observations of the performance trend relative to $\beta$ in Figure 9.
>
> **W2 Would be better to discuss and compare with other methods using signed representations.**
>
> ---
>
> Thank you for the comment.
>
> We compare our SBP method with the signed representation approaches[1][2] suggested in the review on the Cora dataset. As shown, SBP achieves better accuracy than these baselines.
>
> We will also include a discussion of signed representation methods in the Related Work section of the revised manuscript.
>
> | methods | acc |
> | --- | --- |
> | GRAFF[1] | 80.91 |
> | $w$GCN[2] | 78.58 |
> | gat  | 81.48 |
> | Feature-SBP | **82.46** |
> | Label-SBP | **82.90** |
> ---
> [1] Understanding Convolution on Graphs via Energies (TMLR 2023)
>
> [2] Improving Graph Neural Networks with Learnable Propagation Operators (ICML 2023)
>
> **W3 References citation in the appendix.**
>
> ---
>
> Thank you for noting this! We will fix these reference typos in the revision.
>
> **Q1 Clarification.**
>
> ---
> By “non-trivial,” we refer to methods whose negative graphs are constructed via deterministic equations. We provide their mathematical formulations in Table 5 and detail the derivation process in Appendix E.
>
>
> **Q2 Clarification.**
>
> ---
> Regarding $A^+ = A$: here, $A^+$ denotes the positive adjacency matrix used in signed graph propagation, while $A$ refers to the original adjacency matrix of the input graph. Although they may coincide in some cases, conceptually they are distinct.
>
>
>
> **Q3 Label ratio ablation on baselines.**
>
> ---
>
> Thank you for the question.
>
> We conducted a label ratio ablation study for the baselines on the 2-CSBM dataset. As expected, baseline performance improves with increasing label ratio; nevertheless, our SBP consistently outperforms them across all settings.
>
> | train percentage | 20 | 40 | 60 | 80 |
> | --- | --- | --- | --- | --- |
> | batchnorm | 80.25 | 83.00 | 85.50 | 88.25 |
> | contranorm | 83.00 | 86.50 | 86.75 | 90.00 |
> | Label-SBP | **87.00** | **91.50** | **91.50** | **97.50** |
>
> **Q4 Feature-SBP-v2 on ogbn-arxiv.**
>
> ---
>
> Thank you for the question. Here are the Feature-SBP-v2 results on ogbn-arxiv. While it outperforms the vanilla GCN, it underperforms compared to the normalization-based baselines and its label-based counterpart, Label‑SBP‑v2.
>
> |  | L=2 | L=4 | L=8 | L=16 |
> | --- | --- | --- | --- | --- |
> | GCN | 67.32 | 67.79 | **65.54** | 59.13 |
> | Feature-SBP-v2  | **67.89**   | **68.47**  | 65.09 | **60.34** |
>
> **Q5 Ablation on residual parameters.**
>
> ---
>
> Thank you for the question. We present ablation results on the residual parameter $\lambda$ for the CSBM dataset, varying it over {0.1, 0.5, 0.9}. We observe that larger values of $\lambda$ lead to better performance.
>
> |  | 0.1 | 0.5 | 0.9 |
> | --- | --- | --- | --- |
> | Feature-SBP | 79.00 | 83.75 | 88.00 |
> | Label-SBP | 85.75 | 88.25 | 91.50 |
>
> **Q6 Clarification.**
>
> ---
> The notation “#L = 2” in Table 10 is intentional and not a typo; it indicates that the number of layers $L$ is 2.
>
> **Q7 Is SBP applicable to other graph tasks, such as link prediction?**
>
> ---
>
> Thank you for the question.
>
> We have evaluated SBP on the link prediction task, and the results on Cora are provided below. SBP retains high accuracy, demonstrating its applicability beyond node classification.
> | methods | acc |
> | --- | --- |
> | gcn | 0.8389 |
> | Feature-SBP | 0.8430 |
> | Lable-SBP | **0.8617** |
>
> -------
> We appreciate your questions and comments very much. Please let us know for any further questions.

---

> > ### Comment · Reviewer_mzv3 · 2025-08-05
> > **Response to Rebuttal**
> >
> > I thank the authors for their response. I think some of my questions were misunderstood so I would like to ask for clarity on them again, in addition to a few that arose in response to the rebuttal.
> >
> > W2 - Thank you for providing the comparison. However, it would be nice to also compare on a heterophilic dataset.
> >
> > Q2 - I understand the conceptual difference between $A^+$ and $A$. My point was that in the paper they are always considered to be equal. And in this context, my remaining questions in the original Q2 are still unanswered.
> >
> > Q4 - Could there be a possible explanation as to why Feat-SBP-v2 performs much worse than Label-SBP-v2 on the obg-arxiv dataset? Could this be expected as a trend on all larger datasets so that Feat-SBP-v2 is not really as effective?
> >
> > Q5 - How many layers were used for this ablation? Could the same values of lambda also be evaluated without using a signed graph (i.e. only having residual connections in the same GNN backbone as used in SBP), in order to understand the relative contribution of the signed graph component and the residual connections component?
> >
> > Q6 - I understand that '$L=2$' denotes 2 layers. I mean is the value reported for L=2 correct? If so, it has been wrongly highlighted as the second best result in Table 10 in the first column as SGC clearly outperforms Feat-SBP for L=2.

---

> ### Author Response · Authors · 2025-08-08
> **Response to additional questions**
>
> Thank you for your reply.
>
> **W2 would be better to add the results on the heterophilic dataset**
>
> ---
>
> Thank you for your comments! Here, we compare SBP and the baselines on the heterophilic Texas dataset. As shown, SBP achieves better accuracy than these baselines.
>
> | methods | acc |
> | --- | --- |
> | graff[1] | 75.73 |
> | wGCN[2] | 75.68 |
> | GAT | 58.38 |
> | Feature-SBP | **78.38** |
> | Label-SBP | **78.38** |
>
> We will also include these discussions in the Related Work section and the experiments in the revised manuscript.
>
> **Q2 Further clarification and deep analysis of** $A^+=A$
>
> ---
>
> Thank you for your question regarding further clarification and analysis of $A^+ = A$.
>
> >Is  $A^+ = A$ necessary?
>
> $A^+ = A$ is not necessary. In this paper, we summarize nine classic anti-oversmoothing methods (as seen in Table 5) and observe that most of their $A^+$ matrices are either equal to $A$ or are linear combinations of $A$. Following these analyses, in our paper, we set $A^+ = A$ of our SBP and designed the negative adjacency matrix $A^-$ accordingly, to enhance the overall structurally balanced properties for the signed graph propagation.
>
> >The statement on line 135 about beta=0 resulting in the model degenrating into standard unsigned propagation is only true in this case.
>
> We appreciate your observation; Our meaning in the paper is that if $\beta = 0$, there will be no negative edges contributing to the propagation in equation 1, leaving only the positive edges corresponding to attraction, which is indeed similar to the unsigned graph propagation that only considers attraction.
>
> Thank you for your point that if $A^+$ is not equal to $A$, it is not strictly the same. We will clarify this point in our revision; thank you for bringing it to our attention.
>
> >If inter-class edges are removed in Label-SBP-v2, is $A^+ \neq A$?
>
> Thank you for the question. Edges are removed through the negative adjacency $A^-$ (see (4)), and the resulting removal inter-class edges are reflected on the overall signed adjacency matrix $A_s$, so **$ A^+ = A$ still holds in this case.**
>
> >Why are the $\alpha$ values fixed to 1 and not varied?
>
> Both $\alpha$ and $\beta$ are hyperparameters. For simplicity, we fix $\alpha$ and vary the range of $\beta$ from 0.1-100 (refers to accuracy curves versus $\beta$ in Figure 4) to represent changes in the relative strengths of attraction and repulsion.
>
> >Even in the case $A^+ = A$, could it not be possible that the existing graph is not always fully useful [4,5,6] and thus its contribution can be lowered?
>
> We appreciate your insightful suggestion: It is possible that existing graphs may not always be fully useful as suggested by [4, 5, 6], and their contributions can indeed be lower. In this paper, our SBP aims to consider the comprehensive signed adjacency matrix $A_s=\alpha A^+-\beta A^-$, which we assign positive and negative edges, respectively, to mitigate intra-class convergence and thus prevent oversmoothing. The point you mentioned is what we had not previously explored in depth, and we sincerely appreciate your suggestion. We will include a discussion of this aspect of the full use of $A$ in as a potential future direction in our revision.
>
> Thank you once again for your insightful comments.
>
> **Q4 Further analysis of Feature-sbp-v2**
>
> ---
>
> Thank you for your insightful question! Indeed, we found that feature-SBP-v2 is slightly less effective than label-SBP-v2 on large-scale graphs, but not significantly worse. In Table 15, we tested on ogbn-products, and the results are shown below. Notably, feature-SBP-v2 still outperforms vanilla GCN. Therefore, we recommend using label-SBP-v2 for large graphs, as it yields better performance and enhances graph sparsity (Related discussion refers to Appendix I.3).
>
> |  | acc | time |
> | --- | --- | --- |
> | gcn |  73.96  | 00:06:33 |
> | Feature-SBP-v2 |  **74.90**  | 00:06:43 |
> | Label-SBP-v2 | **76.62**  | 00:06:39 |
>
> **Q5 Clarification of Ablation on $\lambda$**
>
> ---
>
> We used a layer count of 200. Below are the results of only using residual connections. We found that the accuracy of the residuals is quite low, indicating that they have already experienced oversmoothing. This suggests that the improvement in accuracy is not due to the residual connections themselves, but rather because of our effective configuration of intra-class positive and inter-class negative edges.
>
> |  | 0.1 | 0.5 | 0.9 |
> | --- | --- | --- | --- |
> | residual | 45.00 | 45.00 | 45.75 |
> | Feature-SBP | **79.00** | **83.75** | **88.00** |
> | Label-SBP | **85.75** | **88.25** | **91.50** |
>
> **Q6**
>
> ---
>
> Thank you for pointing it out! We are sorry for the typo when generating the table. The numbers are correct, we will correct the highlight in the revision.
>
> ---
>
> We appreciate your questions and comments very much. Please let us know for any further questions or adjustments.

---

### Decision · Program_Chairs · 2025-09-17

**Decision:**

Accept (poster)

**Comment:**

This paper tackles the over-smoothing problem in GNNs by introducing a unified "signed graph" perspective, demonstrating that common mitigation strategies are equivalent to implicitly adding negative edges and inducing an "attraction-repulsion" mechanism. The authors emphasize that the structured organization of positive and negative edges is key, rather than their mere presence. To this end, they propose Structural Balanced Propagation (SBP), a parameter-free, plug-and-play method grounded in structural balance theory, with both label-based and feature-based variants. Theoretical analysis and experiments on nine datasets, including deep (300-layer) models, show SBP’s superiority over traditional approaches.

Reviewers unanimously recognized the paper’s valuable and novel theoretical contributions and recommended acceptance, while noting some limitations such as strong assumptions and relatively limited large-scale experiments. The AC concurs with the reviewers and recommends acceptance.